# Behavioral consequences of second-person pronouns in written communications between authors and reviewers of scientific papers

Zhuanlan Sun[1,4], C. Clark Cao [2,4], Sheng Liu[2,4], Yiwei Li [2] ✉ & Chao Ma [3] ✉

Pronoun usage's psychological underpinning and behavioral consequence have fascinated researchers, with much research attention paid to second-person pronouns like "you," "your," and "yours." While these pronouns' effects are understood in many contexts, their role in bilateral, dynamic conversations (especially those outside of close relationships) remains less explored. This research attempts to bridge this gap by examining 25,679 instances of peer review correspondence with *Nature Communications* using the difference-in-differences method. Here we show that authors addressing reviewers using second-person pronouns receive fewer questions, shorter responses, and more positive feedback. Further analyses suggest that this shift in the review process occurs because "you" (vs. non-"you") usage creates a more personal and engaging conversation. Employing the peer review process of scientific papers as a backdrop, this research reveals the behavioral and psychological effects that second-person pronouns have in interactive written communications.

In written communications, one can address the other conversational party using either second-person pronouns or their third-person counterparts. For instance, during the peer review process of a scientific paper, an academic may address the reviewers either using "you" (e.g., "the issue you brought up") or a third-person reference instead (e.g., "the issue the reviewer brought up…"). Whether this choice matters, however, is less known. This question is embedded within the recent research investigating the behavioral and psychological consequences of personal pronoun usage[1–3], which in turn falls under the broader research category of the social function of language usage[4–6]. Building upon this growing literature, the present research aims to investigate how the usage of second-person pronouns ("you," "your," and "yours"; hereinafter, we use the terms second-person pronoun usage and "you" usage interchangeably) impacts the outcome of written communications.

Currently, a wealth of research has investigated the impact of "you" usage on individuals' mental state and/or behavior. For instance, "you" can draw the attention of a conversational party and hence evoke higher involvement[7–9]. Moreover, generic "you," as in "you shall not murder," signals normative behavior and hence impacts persuasion[10–13]. Furthermore, "you" usage in lyrics like "I will always love you" or movie quotes like "here's looking at you, kid" can remind one of somebody in their own life (a loved one in these examples)[14]. Despite their important insights, however, most such investigations focus on one-way and one-off communications. While another body of literature does investigate "you" in two-way communications, it is largely limited to close relationships, mainly focusing on how pronoun usage reflects a party's self- or other-focus[4,15–17]. Therefore, the field's knowledge is still limited about the role of second-person pronouns in bilateral, dynamic, and interactive conversations, especially beyond close relationships.

[1]High-Quality Development Evaluation Institute, Nanjing University of Posts and Telecommunications, Nanjing, China. [2]Department of Marketing and International Business, Lingnan University, Hong Kong, China. [3]School of Economics and Management, Southeast University, Nanjing, China. [4]These authors contributed equally: Zhuanlan Sun, C. Clark Cao, Sheng Liu. ✉e-mail: victor.li@ln.edu.hk; machao@seu.edu.cn

To bridge this gap, in the present paper, we examine the behavioral and psychological consequences of second-person pronoun usage in interactive, conversational settings. Specifically, by analyzing 25,679 instances of revision correspondence with *Nature Communications*, we focus on how "you" (vs. non-"you") usage in authors' responses to reviewers may influence reviewers' behavior. This dataset is ideal for our investigation, because the peer review process allows us to compare naturally occurring instances of both "you" and non-"you" responses.

The extant literature has shown that by directly addressing a conversational party, second-person pronouns can evoke the listener's attention, personal relevance, and involvement in the communication[7,14]. Other personal pronouns do not possess this feature. For instance, in stark contrast to "you," third-person pronouns often function to signal objectivity and minimize the involvement or even the existence of the speaker[18–20]. Building on this literature, we contend that in a communicative setting, addressing the other party as "you" (vs. not as "you") should be associated with a more personal and engaging conversation, in contrast to an impersonal, businesslike exchange.

This feature of "you" usage may, in turn, lead to observable behavioral patterns in peer review outcomes. First, the personal and engaging conversational tone stimulated by "you" usage may in and of itself make the reviewer like the responses more, as individuals tend to favor things that are personally relevant[8,14]. Second, communicative norms that govern such conversations may call for greater politeness, civility, and embarrassment avoidance ("face-saving") in communications[21–24], making the comments more favorable (or less harsh) than they otherwise would be and resulting in greater positivity and fewer questions in reviewer comments.

Building on this perspective, here we show that when the authors use (vs. do not use) second-person pronouns to address the reviewers, they also see less lengthy reviewer comments, encounter fewer questions, and receive more positive and less negative feedback. We further link this shift in the review process to a more personal and engaging conversation prompted by "you" usage: First, when authors address reviewers using "you," the reviewer responses tend to include fewer first-person singular pronouns, suggesting decreased self-focus[25–28]; and to use less complex words, a staple feature of in-person conversation[29–32]. Second, thematic analyses conducted using Latent Dirichlet Allocation (LDA) show that second-person pronouns are indeed associated with increased reviewer engagement in their comments. Core findings from our dataset are also causally supported by a pre-registered behavioral experiment ($N = 1601$). Specifically, when participants assuming the role of reviewers are addressed in second person (vs. third person), they evaluate an otherwise identical author response as more positive. This effect is mediated by the extent to which the conversation is perceived as personal and engaging. Taken together, this research investigates the behavioral consequence and

psychological underpinning of second-person pronoun usage employing field and lab data. In so doing, we contribute to the literature on language usage (and pronoun usage in particular) and shed light on the collegiate understanding of the peer review process and science of science in general.

## Results
### Data and design
We analyzed revision correspondence of all papers published in *Nature Communications* between April 2016 (when the journal first began publishing reviewer reports) and April 2021. This dataset contains 13,359 published papers that account for a total of 29,144 rounds of review. In the present research, a "round" of review is defined as one exchange between the editorial/reviewer team (hereinafter simply "reviewers") and the authors, with the reviewer comments being followed by the author responses. For instance, the "1st round of review" begins with the initial comments from the reviewers and the authors' responses to those comments, the 2nd round of review consists of the next batch of reviewer comments and the authors' responses to them, and so on. In our analysis, we focus on the authors' usage of second-person pronouns in addressing the reviewer team in the first review-response-review process (i.e., reviewer comments in the 1st round, author responses in the 1st round, and reviewer comments in the 2nd round). We focus on this process because it constitutes most of our observations (25,679, or 88.11% of 29,144 review rounds) and, more importantly, affords a difference-in-differences (DID) design, which we elaborate below. Figure 1 illustrates our focal data and study design (full details regarding the number of papers and rounds of review can be found in Supplementary Note 1).

In examining the impact of "you" usage, it is important to consider some distinctive features of the peer review process. As an illustration, Fig. 2 shows the authors' response to the reviewers in the 1st round of review (marked by the vertical dashed line), as well as the number of questions reviewers posed before and after this response (i.e., question counts in the 1st and 2nd review rounds). We then compare question counts following both "you" and non-"you" usage in a quasi-experimental fashion. Specifically, we categorize a paper into the treatment group if its authors used "you" in their 1st-round responses (which, in our context, can be considered as the treatment administered to reviewers), and the control group if they did not. Note that the treatment and control groups here are not in the strict experimental sense, as the papers are not randomly assigned to them.

Importantly, as illustrated in Fig. 2, to estimate the effect of "you" usage, it could be misleading to simply contrast the number of questions reviewers raised after the author responses with "you" (5.82) and without (4.25). This is because empirically, the treatment and control groups may begin with different question counts (which happens to be the case in our data−29.87 and 24.30, respectively, as per Fig. 2). To offset this initial discrepancy, we instead measure the decline in

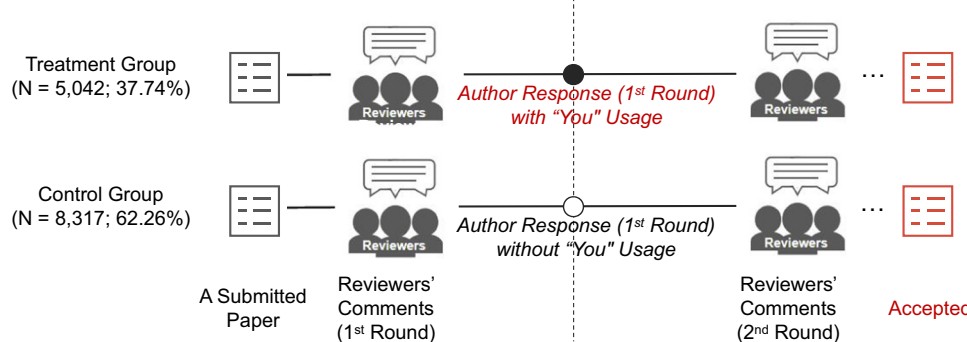

**Fig. 1 | The review process, rounds, and paper grouping.** The treatment group and control group are defined by whether the authors responded to reviewers' comments using second-person pronouns in the 1st round of review (i.e., between the 1st and 2nd round reviewer comments).

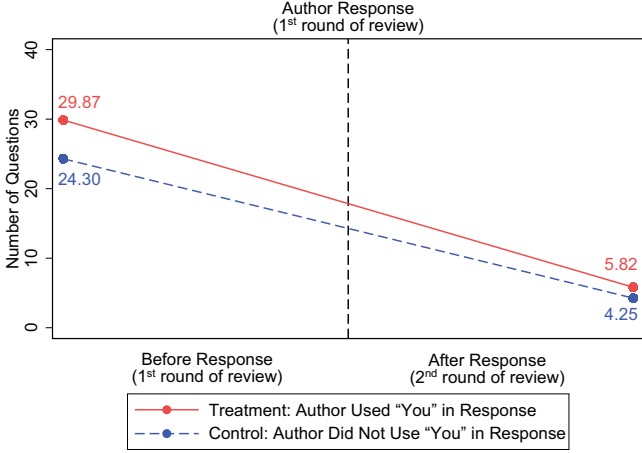

**Fig. 2 | Number of questions in reviewer comments (1st and 2nd rounds).**
Numbers herein represent mean values in the raw data and are not adjusted for any control variables or fixed effects. Supplementary Fig. 1 illustrates the distribution of the number of questions (and of words) across four scenarios as yielded by the (author "you" usage: yes vs. no) x 2 (round of review: before vs. after author response) DID design. We also direct interested readers to Supplementary Note 2 for further insights into data patterns related to "you" usage.

question count from the 1st to the 2nd round. Specifically, authors who used "you" saw a decrease of 24.05 questions in the subsequent round, while those who did not use "you" saw a decrease of 20.05.

Here, the control group reduction of 20.05 questions reflects a "natural" progression in our data, such that question count dwindles as the review process progresses, regardless of whether the author used "you" (see Fig. 2). On the other hand, the treatment group reduction of 24.05 also encompasses our focal effect of "you" usage, in addition to the overall trend. Therefore, the difference between the two reductions provides a relatively precise estimation of the effect of "you" usage. Specifically, compared to the control group, the treatment group experienced a steeper decline in question counts (by a margin of 4).

This approach to estimating the effect of "you" usage constitutes a DID framework, a quasi-experimental method widely used in observational data analysis. Specifically, the first "differences" here are the differences in question counts before and after author response, within both the treatment and control groups. These differences serve to offset initial discrepancies between the groups. The second "difference" then contrasts these two differences to estimate the effect of "you" usage—hence the name "difference-in-differences."

As depicted in Fig. 1, during the 1st round of review, authors of 5042 papers (37.74% of all 13,359 papers) used "you" in their responses to the reviewer comments (i.e., treatment group), whereas authors of 8317 (62.26%) papers did not use "you" (i.e., the control group). We then estimate the effect of "you" on various behavioral and psychological outcomes by comparing the average change in such outcomes before and after a response with "you" versus without "you". In what follows, this DID model enables us to examine more closely the effect of "you" usage on total number of questions from the reviewers, total length of reviewer comments, and positivity/negativity of the 2nd-round reviewer comments.

In addition, we also employ this DID model to examine the impact of "you" usage on how personal and engaging the reviewer-author communication is. Measurements of interest include the subjectivity of the language used in the reviewer comments, the frequency of reviewers' use of first-person pronouns, the complexity of the vocabulary used by the reviewers, and the extent to which the reviewers engage with the authors.

Equation (2) in the Methods section formulates the DID model summarized above. To ensure the robustness of our analysis, a variety of control variables are also included in this model. The summary statistics of our dependent and control variables are presented in Table 1.

### Reviewers wrote less and asked fewer questions following authors' "you" (vs. non-"you") usage

Table 2 summarizes the DID estimates on two review outcomes: the total number of questions the reviewer raised, and the total number of words the reviewer wrote. The first non-header row is of particular interest, as it reports the DID estimator, or the effect of the interaction between "you" usage and time (i.e., before vs. after the author's response).

In Column (1), the significant, negative coefficient (−4.0019) indicates that "you" usage has a negative effect on the total number of questions the reviewer asked. Specifically, when exposed to "you" (vs. non-"you") author response in the 1st round of review, reviewers raised fewer questions in the 2nd round ($t(25675) = -10.01$, $p < 0.001$, B = −4.00, 95% CI = [−4.79, −3.22]). This result remains robust when the control variables (see Table 1) and paper fixed effects are included in the DID model ("you" usage sees 3.34 fewer questions; $t(12319) = -9.40$, $p < 0.001$, B = −3.34, 95% CI = [−4.03, −2.64]; Column (2)). Similarly, reviewers addressed by "you" (vs. non-"you") language also wrote 172.15 fewer words as estimated by the basic DID model ($t(25675) = -9.54$, $p < 0.001$, B = −172.15, 95% CI = [−207.50, −136.79]; Column (3)), or 135.59 fewer words when the control variables and paper fixed effects are included ($t(12319) = -9.36$, $p < 0.001$, B = −135.59, 95% CI = [−163.98, −107.20]; Column (4)).

### Reviewer Comments Are More Positive (and Less Negative) Following Authors' "You" (vs. Non-"You") Usage

In addition, we find that authors using "you" also receive more positive (and less negative) reviewer comments during the review process. To assess positivity in a reliable and robust manner, we employed multiple widely-adopted automated text analysis techniques to analyze the reviewer comments (see Sentiments of Reviewers' Comments in the Methods section for more details on these measurements).

Table 3 summarizes the corresponding DID estimates, with control variables and paper fixed effects included. Columns (1) and (2) reflect the positivity of reviewer comments employing the Python package TextBlob and R package sentimentr, respectively. Columns (3) and (4), on the other hand, assess the negativity of reviewer comments employing the Python package NLTK and a hand-coded lexicon of common negative words, respectively. As indicated in the first non-header row of Table 3, the findings are consistent across all measurements of review positivity/negativity, such that authors' use of "you" in the 1st round is significantly associated with increased positivity and decreased negativity of the reviewer comments in the 2nd round.

To further validate these findings, we conducted six additional robustness checks, the detailed results of which are reported in the Supplementary Information for succinctness. To briefly summarize: First, we demonstrate that more "you" usage is associated with a stronger effect on the variables above (Supplementary Note 4 and Supplementary Table 3). Second, to construct a cleaner treatment group, we included a paper in the treatment group only when its "you" usage is conversational (as opposed to courteous; e.g., "thank you"). Of all 5042 "you" papers, 1847 (36.63%) contain only courteous "you," and are thus excluded from analysis during this robustness check (Supplementary Note 4 and Supplementary Table 4). Third, to construct a cleaner control group, we only included a paper in the control condition if it explicitly addresses the reviewer in third person (e.g., "the reviewer"; Supplementary Note 4 and Supplementary Table 5). Fourth, we employed a matching technique (propensity score matching, PSM) to obtain matched treatment and control groups with comparable

**Table 1 | Summary statistics of dependent and control variables**

| Variable Names | Measurement | Mean/ Percentage | SD | Min | Max |
|---|---|---|---|---|---|
| Dependent Variable | | | | | |
| Number of Questions | Number of questions raised by reviewers. | 16.070 | 18.758 | 0.000 | 435.000 |
| Number of Words | Number of words contained in a reviewer report. | 1069.450 | 925.370 | 1.000 | 5577.000 |
| Positivity (Python) | Reviewers' positive sentiment measured by Python package TextBlob on a scale ranging from –1 (very negative) to 1 (very positive). | 0.124 | 0.097 | –0.383 | 1.000 |
| Positivity (R) | Reviewers' positive sentiment measured by R package sentimentr on a scale ranging from –1 (very negative) to 1 (very positive). | 0.131 | 0.154 | –0.850 | 0.998 |
| Negativity (Python) | Reviewers' negative sentiment measured by Python package NLTK on a scale ranging from 0 (not negative at all) to 1 (very negative). | 0.028 | 0.031 | 0.000 | 0.524 |
| Negativity (Hand Coded) | Frequency of hand-coded negative words contained in reviewers' comments on a custom negativity scale that starts at 0 (not negative at all) and increases by 0.01 (or 1%) each time one of the 92 common negative words from a hand-coded lexicon appears. | 0.026 | 0.034 | 0.000 | 0.430 |
| Subjectivity | Reviewers' subjective sentiment measured by Python package TextBlob on a scale ranging from 0 (not subjective at all) to 1 (very subjective). | 0.477 | 0.095 | 0.000 | 1.000 |
| First-person Singular Pronoun Usage | Number of first-person singular pronouns contained in a reviewer report. | 8.102 | 8.323 | 0.000 | 127.000 |
| Word Complexity | Average number of syllables in words in a reviewer report. | 1.954 | 0.133 | 0.603 | 9.669 |
| Reviewer Engagement | Level of the reviewer engagement calculated by Eq. (1). | 1.193 | 0.988 | –4.807 | 7.330 |
| Control Variable | | | | | |
| Number of Pages | Number of pages in a paper. | 11.250 | 3.121 | 5.000 | 24.000 |
| Number of References | Number of references in a paper. | 56.556 | 16.012 | 1.000 | 212.000 |
| Title Length | Number of words contained in the title of a paper. | 11.418 | 2.804 | 1.000 | 21.000 |
| Number of Authors | Number of authors of a paper. | 10.559 | 13.278 | 1.000 | 548.000 |
| H-index of the First Author | The H-index of the first author supplemented from the Web of Science Database. | 14.731 | 12.300 | 0.000 | 155.000 |
| Gender of the First Author | Dummy variable that takes a value of 1 if the gender of the first author of a paper is female. | 0.306 | 0.461 | 0.000 | 1.000 |
| Last Initial of the First Author[a] | 26 dummy variables indicating the last initial of the first author. | NA | NA | NA | NA |
| Positivity of Authors (1st Round) | Authors' positive sentiment of 1st review round measured by Python package TextBlob. | 0.087 | 0.044 | –0.750 | 1.000 |
| Friendliness of Authors (1st Round) | Frequency of hand-coded friendly words used by authors during the 1st review round process. | 0.279 | 0.214 | 0.000 | 3.900 |
| Positivity of Reviewers (1st Round) | Reviewers' positive sentiment of 1st review round measured by Python package TextBlob. | 0.099 | 0.038 | –0.171 | 0.344 |
| Month of Publication[a] | 12 dummy variables indicating month of publication. | NA | NA | NA | NA |
| Publication Year[a] | 6 dummy variables indicating year of publication (2016–2021). | NA | NA | NA | NA |
| Paper Discipline[a] | 5 dummy variables indicating the discipline of the paper (biological sciences, physical sciences, health sciences, earth and environmental sciences, scientific community and society). | NA | NA | NA | NA |

[a]Refer to Supplementary Note 3 and Supplementary Table 2 for detailed levels and data distributions of variable.

observable characteristics (Supplementary Note 4; Supplementary Tables 6, 7; Supplementary Fig. 3). Fifth, we employed a two-stage Heckman model (Supplementary Note 4; Supplementary Tables 8, 9) to capture authors' "you" usage in response to the initial use of "you" by reviewers. In doing so, we allow for a more reciprocal, dynamic view of the impact of "you." Sixth, to further account for the non-randomness in "you" usage, we employed placebo (non-parametric permutation) tests to validate that our DID findings are not spurious (Supplementary Note 4 and Supplementary Fig. 4). Our results remain robust to all these robust checks.

**More personal and engaging conversation following "you" (vs. non-"you") usage: "I" usage, word complexity, and reviewer engagement**

Thus far, we have demonstrated that addressing reviewers as "you" is associated with fewer questions and less writing from the reviewers, as well as more positive and less negative reviewer comments. In postulating the underlying mechanism of the effect, we contend that addressing the other party in second person is also associated with a

more personal and engaging conversation, which is in turn responsible for these marked effects on the review process. Below, we examine several potential indicators of personal and engaging conversations to test this hypothesis employing the same DID model.

A potential indicator is the subjectivity of reviewer comments—the extent to which comments reflect personal opinions rather than factual information[33]. High subjectivity in language indicates that the text or utterance is more opinionated and personal (as opposed to factual and unbiased) in nature. The usage of subjective languages is a marked feature of interpersonal conversations[34–37]. We assess language subjectivity using the Python package TextBlob[38] (see Supplementary Note 5 for method details, and Supplementary Fig. 5 for the most frequently used words in our data indicating subjectivity). However, our prediction that authors' "you" usage is associated with increased subjectivity in reviewer responses does not reach the 0.05 level of significance (see Column (1) in Table 4).

One evidence of a more personal and engaging conversation involves first-person pronoun usage. In our data, authors' "you" usage is associated with reviewers' decreased usage of first-person singular

pronouns (e.g., "I," "me," "my"; see Column (2) in Table 4), which can indicate self-focused attention[25–28]. On the other hand, there is no statistically significant difference for first-person plural pronouns (i.e., "we," "us," "our"), which often indicate a communal focus[39] (Supplementary Note 6 and Supplementary Table 12). This result suggests that following authors' "you" usage, reviewer may show less self-focus, hence making fewer "I" statements.

Additional evidence of a more personal conversation is found in word complexity. A reviewer comment is more complex if the words in it contain more syllables on average. We find that authors' second-person pronoun usage is associated with decreased word complexity

in reviewer comments (see Column (3) in Table 4). This result suggests that the reviewers, when addressed using second-person pronouns, favored more plain, readable language over complex and formal written language, a choice often made to facilitate a conversation[29–31,40,41].

Yet more evidence of personal, engaging conversation is found by employing the text mining technology Latent Dirichlet Allocation (LDA), which identifies the hidden topics in reviewer comments that may potentially indicate reviewers' engagement with the authors. R package topicmodels was applied on reviewer comments of 1st and 2nd round, and revealed 40 hidden topics at optimal best model fit

**Table 2 | Reviewers raised fewer questions and wrote fewer words following authors' "you" (vs. non-"you") usage**

|  | (1) Number of Questions | | (2) Number of Questions | | (3) Number of Words | | (4) Number of Words | |
|---|---|---|---|---|---|---|---|---|
|  | Coef. | t-value | Coef. | t-value | Coef. | t-value | Coef. | t-value |
| Response with "You" | −4.0019*** | −10.01 | −3.3360*** | −9.40 | −172.1471*** | −9.54 | −135.5879*** | −9.36 |
| × After Response | (0.3998) |  | (0.3550) |  | (18.0369) |  | (14.4849) |  |
| Response with "You" | 5.5687*** | 15.30 | NA | NA | 269.8819*** | 17.78 | NA | NA |
|  | (0.3639) |  | (NA) |  | (15.1767) |  | (NA) |  |
| After Response | −20.0423*** | −92.84 | −21.4706*** | −108.28 | −1148.0390*** | −112.09 | −1220.0409*** | −142.94 |
|  | (0.2159) |  | (0.1983) |  | (10.2419) |  | (8.5355) |  |
| Control Variables | No |  | Yes |  | No |  | Yes |  |
| Paper Fixed Effects | No |  | Yes |  | No |  | Yes |  |
| Observations | 25,679 |  | 24,640 |  | 25,679 |  | 24,640 |  |
| R² | 0.341 |  | 0.523 |  | 0.440 |  | 0.662 |  |

Each column in the table represents a DID regression; Columns (1) and (3) are basic settings without controls, and columns (2) and (4) include control variables and paper fixed effects. NA indicates that the pure effect of "you" usage is absorbed by paper fixed effects. Standard errors are reported in parentheses and are clustered at the paper level for columns (2) and (4). *$p < 0.05$, **$p < 0.01$, ***$p < 0.001$. Two-sided $t$ tests with a 95% confidence interval are employed here and throughout the paper.

**Table 3 | Reviewer comments become more positive and less negative following authors' "you" (vs. non-"you") usage**

|  | (1) Positivity (Python) | | (2) Positivity (R) | | (3) Negativity (Python) | | (4) Negativity (Hand Coded) | |
|---|---|---|---|---|---|---|---|---|
|  | Coef. | t-value | Coef. | t-value | Coef. | t-value | Coef. | t-value |
| Response with "You" | 0.0051* | 2.12 | 0.0173*** | 4.45 | −0.0019*** | −2.88 | −0.0048*** | −6.83 |
| × After Response | (0.0024) |  | (0.0039) |  | (0.0007) |  | (0.0007) |  |
| After Response | 0.0510*** | 34.56 | 0.0679*** | 28.45 | −0.0264*** | −59.68 | −0.0317*** | −78.10 |
| Control Variables | Yes |  | Yes |  | Yes |  | Yes |  |
| Paper Fixed Effects | Yes |  | Yes |  | Yes |  | Yes |  |
| Observations | 24,640 |  | 24,640 |  | 24,640 |  | 24,640 |  |
| R² | 0.128 |  | 0.110 |  | 0.288 |  | 0.411 |  |

Each column in the table represents a DID regression with control variables and paper fixed effects. The coefficients of the variable Response with "You" are not included in this table, in that the pure effect of "you" usage is absorbed by paper fixed effects. Standard errors in parentheses are clustered at the paper level. *$p < 0.05$, **$p < 0.01$, ***$p < 0.001$.

**Table 4 | Reviewer comments see fewer "I" usage, use less complex words, and are more engaging following authors' "you" (vs. non-"you") usage**

|  | (1) Subjectivity | | (2) First-person Singular Pronoun Usage | | (3) Word Complexity | | (4) Reviewer Engagement | |
|---|---|---|---|---|---|---|---|---|
|  | Coef. | t-value | Coef. | t-value | Coef. | t-value | Coef. | t-value |
| Response with "You" | 0.0042 | 1.74 | −1.0918*** | −6.84 | −0.0139*** | −4.49 | 0.1078*** | 5.85 |
| × After Response | (0.0024) |  | (0.1596) |  | (0.0031) |  | (0.0184) |  |
| After Response | 0.0194*** | 12.62 | −2.9931*** | −35.56 | 0.0445*** | 22.64 | −0.7525*** | −63.69 |
|  | (0.0015) |  | (0.0842) |  | (0.0020) |  | (0.0118) |  |
| Control Variables | YES |  | YES |  | YES |  | YES |  |
| Paper Fixed Effects | YES |  | YES |  | YES |  | YES |  |
| Observations | 24,640 |  | 24,640 |  | 24,640 |  | 24,640 |  |
| R² | 0.051 |  | 0.522 |  | 0.207 |  | 0.486 |  |

Each column in the table represents a DID regression with control variables and paper fixed effects. The coefficients of the variable Response with "You" are not included in this table, in that the pure effect of "you" usage is absorbed by paper fixed effects. Standard errors in parentheses are clustered at the paper level. *$p < 0.05$, **$p < 0.01$, ***$p < 0.001$.

(see Supplementary Note 7 and Supplementary Fig. 6). Here, we focus on one topic (topic 11) that consists of numerous words reflecting communication and engagement during the review process (Fig. 3). All 40 identified topics and their top 10 marker words are displayed in Supplementary Note 7 and Supplementary Fig. 7. The distribution of document-level topic proportions within the chosen topic (i.e., reviewer engagement) is presented in Supplementary Note 7 and Supplementary Fig. 8.

We measure the engagement level of a review comment by the frequency of words associated with the identified engagement topic. This frequency is equal to the probability of a review containing the engagement topic, multiplied by the total word count of said review. We find that the topic of engagement appears significantly more frequently in reviewer comments if "you" was used (vs. not used) by the authors. This result again suggests that the use of "you" may have triggered greater engagement in the subject matter of the paper (see Column (4) in Table 4). We further experimented with alternative LDA models with 35 and 45 topics, as well as the use of a subset of "high-engagement" words (e.g., "exciting," "interesting," and "enjoy"; see all

116 words in Supplementary Note 8 and Supplementary Table 15) and the adoption of a structural topic model[42,43]. In all alternative models, we obtain results consistent with our predictions (see Supplementary Note 7 and Supplementary Table 13). However, no statistically significant difference was observed between the treatment and control groups when the selected topic of reviewer engagement was replaced with other, unrelated topics (see Supplementary Note 7 and Supplementary Table 14).

## Effect of the usage of second-person pronouns by the reviewers on engagement measurements

If an author's "you" usage can render conversations more personal and engaging, it follows that this effect should grow even stronger when both parties employ "you" to address each other. In this section, we examine how both parties' "you" usage jointly impacts indicators of a personal and engaging conversation. This addition of reviewer usage of "you" into our analyses yields a difference-in-difference-in-differences (DDD) model. This DDD model is best viewed as splitting our original DID model into two separate yet comparable DIDs: one with reviewers who used "you" in the 1st round and the other without. This design thus allows us to examine the impact of reviewers' "you" usage by contrasting the two separate DID models. Indeed, as demonstrated in Supplementary Note 9, Supplementary Tables 16, 17, we find that when the reviewer initiates a "you" (vs. non-"you") conversation in the first place, most of our DID (save for subjectivity) yields a larger effect size. In other words, the effect of "you" usage is the most evident when both parties use "you" language. Table 5 formally compares the effects of the two DIDs, forming a third differential impact based on reviewers' initial "you" usage. The spirit of our analysis echoes that of Kenny and colleagues' seminal work on dyadic data analysis, which factors the role of both parties into the analysis[44,45].

Recall that author's usage of "you" is sufficient to elicit significant behavioral consequences (i.e., question numbers, word counts, positivity, negativity), irrespective of whether the reviewer used "you" first or not (refer to Supplementary Note 9; Supplementary Tables 18, 19). What we attempt to demonstrate here is the amplifying effect of mutual "you" usage on our mechanism—that is, creating a personal, engaging conversation.

Note that although the focus of this research lies in authors' "you" usage, our DDD model, together with the previously discussed Heckman Model, affords a reciprocal perspective into how reviewers' "you"

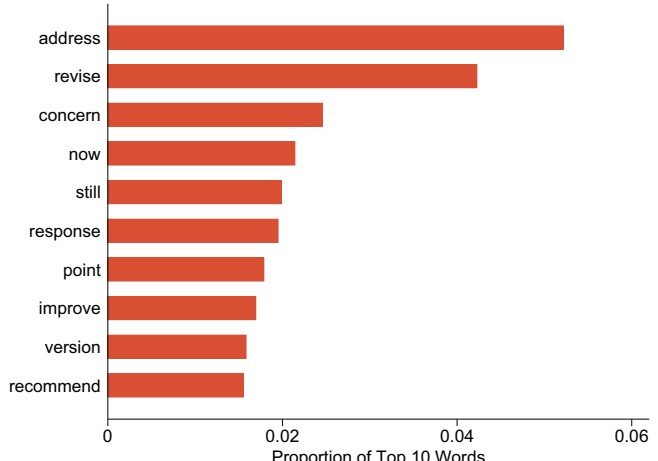

**Fig. 3 | Top 10 highest-probability words in the LDA-identified engagement topic (topic 11).** Clustered into the engagement topic are words such as "address," "revise," "concern," and so on.

**Table 5 | DDD estimates for a "you" conversation initiated by reviewers**

| | (1) Subjectivity | | (2) First-person Singular Pronoun Usage | | (3) Word Complexity | | (4) Reviewer Engagement | |
|---|---|---|---|---|---|---|---|---|
| | Coef. | t-value | Coef. | t-value | Coef. | t-value | Coef. | t-value |
| Response with "You" × After Response | 0.0025 | 0.51 | −1.2022*** | −3.38 | −0.0143* | −2.31 | 0.0803* | 2.07 |
| × "You" Initiated by Reviewers | (0.0050) | | (0.3554) | | (0.0062) | | (0.0388) | |
| After Response | 0.0176*** | 9.51 | −2.0787*** | −24.50 | 0.0486*** | 19.84 | −0.7693*** | −55.66 |
| | (0.0018) | | (0.0849) | | (0.0024) | | (0.0138) | |
| After Response | 0.0067* | 2.05 | −3.4050*** | −15.57 | −0.0151*** | −3.89 | 0.0628* | 2.36 |
| × "You" Initiated by Reviewers | (0.0033) | | (0.2187) | | (0.0039) | | (0.0266) | |
| Response with "You" | 0.0019 | 0.61 | 0.0336 | 0.22 | −0.0049 | −1.25 | 0.0613* | 2.56 |
| × After Response | (0.0032) | | (0.1541) | | (0.0039) | | (0.0239) | |
| Control Variables | YES | | YES | | YES | | YES | |
| Paper Fixed Effects | YES | | YES | | YES | | YES | |
| Observations | 24,640 | | 24,640 | | 24,640 | | 24,640 | |
| R² | 0.052 | | 0.546 | | 0.210 | | 0.487 | |

Each column in the table represents a DDD regression with control variables and paper fixed effects. The coefficients of variables "You" Initiated by Reviewers, Response with "You", and their interactions are not included in this table, in that the pure effect of these variables is absorbed by paper fixed effects. Standard errors in parentheses are clustered at the paper level. *$p < 0.05$, **$p < 0.01$, ***$p < 0.001$.

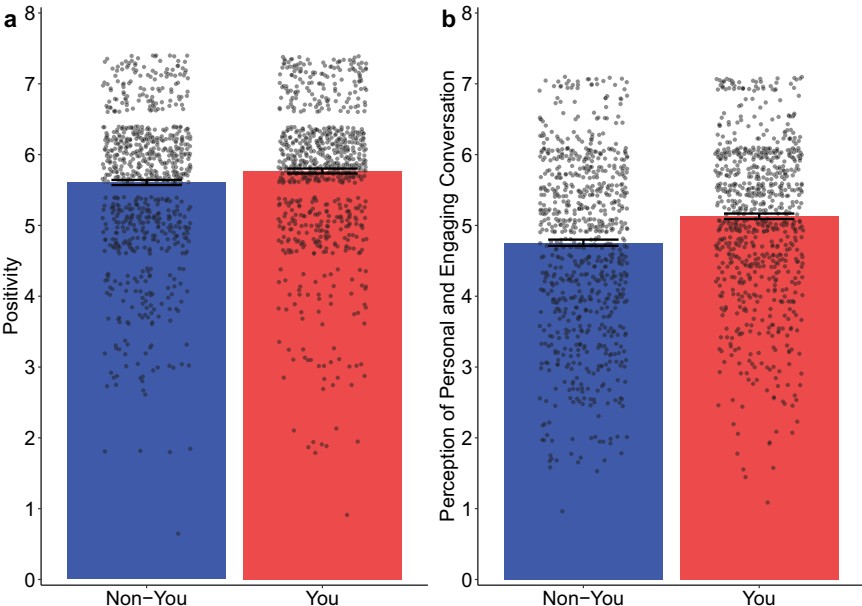

**Fig. 4 | Results of main text experiment. a** Participants addressed with "you" (vs. non-"you") rated the author's response more positively. **b** Participants addressed with "you" (vs. non-"you") found their conversation with the author more personal and engaging. Individual data points are shown using overlaid dot plots. Error bar shows ±1 SE.

usage also impacts the author. Specifically, reviewers' "you" usage can not only stimulate authors' "you" usage (Heckman Model) but also strengthen the contribution of "you" usage to boosted engagement (DDD). Also note that our DDD analysis can also cascade into the remaining rounds, and we direct interested readers to Supplementary Note 2 for more information.

Additionally, we report DDD results for number of questions, number of words, positivity, and negativity in Supplementary Tables 18, 19. Although the DID effect sizes are generally larger when reviewers used "you" in the 1st round, these DDD results are not statistically significant.

### Behavioral experiment

The above analyses provide converging evidence that "you" usage is associated with more personal and engaging communication. However, secondary data have a limited capacity for establishing psychological mechanisms and, crucially, causality. To address this, we conducted a controlled, pre-registered (https://aspredicted.org/9yw2f.pdf) experiment to supplement our field data. In this study, 1601 participants were asked to play the role of reviewers and evaluate an author's response. Of all participants, 901 (56.3%) self-identified as female, 676 (42.2%) as male, and 24 (1.5%) as non-binary or chose not to disclose their gender; $M_{age}$ = 41.9 years.

Participants were randomly assigned to one of two conditions, in which they were addressed by the author using either "you" or non-"you" language. Participants then responded to a battery of questions regarding the author's response. Detailed design and procedures are outlined in the Methods section. Key findings are summarized below, while secondary analyses are available in Supplementary Method 1.

First, an ANOVA reveals that participants addressed with "you" rated the author's response more positively ($M$ = 5.77, SD = 0.98) than did those who were not (M = 5.61, SD = 1.01; $F(1, 1599)$ = 10.62, $p$ = 0.001, Cohen's $d$ = 0.16, 95% CI = [0.06, 0.26]). Furthermore, "you" (vs. non-"you") usage also led participants to perceive their exchange with the author as more personal and engaging ($M$ = 5.13, SD = 1.10 vs. $M$ = 4.76, SD = 1.24; $F(1, 1599)$ = 40.78, $p$ < 0.001, Cohen's $d$ = 0.32, 95% CI = [0.22, 0.42]). Figure 4 illustrates these findings.

Second, a mediation analysis shows that the relationship between "you" usage and positivity is fully mediated by participants' perception of an personal and engaging communication (unstandardized indirect effect = 0.19, SE = 0.03, 95% CI = [0.14, 0.26]; 5000 bootstrap resamples).

Taken together, "you" usage indeed makes the reviewer-author communication more personal and engaging, which in turn leads to more positive reviewer comments. To further validate these results, we also replicated the main effects and the mediation effect above in a separate sample ($N$ = 1200) employing the same experimental design. In this second experiment, we also find that these findings cannot be attributed to alternative processes such as contention, personal connection, or obligation. Refer to Supplementary Method 2 for detailed results.

## Discussion

This work examines the correspondence in the peer review process and finds that when author responses use (vs. do not use) second-person pronouns (e.g., "you"), reviewers ask fewer questions, provide briefer responses, and offer more positive and fewer negative comments. Both lab and field evidence converge to demonstrate that this is the case because "you" (vs. non-"you") usage fosters a more personal and engaging conversation.

An apparent practical implication of this work is, of course, that authors of academic papers can employ second-person pronouns strategically during the review process to their benefit. However, we believe that our findings extend beyond academic contexts and could be relevant for other forms of (formal) written communication. For example, businesses might utilize "you" in their marketing materials to nudge consumer attitude; likewise, professionals or politicians could use "you" to foster greater engagement. While the effectiveness of these applications requires further empirical validation, the real-world implications of our findings prove both intriguing and potentially impactful.

Conceptually, our study first contributes to the broad literature on language usage, particularly pronoun usage. Researchers have long known that nuances in language use matter. For example, the presence or absence of future tense in a language affects its users' future orientation[6], and word choice can signal political stance[46]. Within this

field, pronoun usage has fascinated theorists for decades, as it can reflect individuals' mental states such as narcissism[27] or lead to various mental processes or behaviors (such as introducing independence/interdependence self-construal)[47,48]. Recent technological advancements have significantly fueled research on pronoun usage, enabling the collection of large amounts of data from various online platforms[49–51].

With respect to second-person pronouns, while their usage has been studied in unidirectional, one-off communication[7–14], understanding "you" usage in dynamic, bilateral, reciprocal contexts remains critical. Thus far, important work has explored the bilateral usage of "you" in close relationships[4,15–17]. Additionally, methods like the Actor-Partner Interdependence Model have further enriched our understanding of communications between comparable parties[44,45]. Nonetheless, current insights into mutual "you" usage are mostly confined to close relationships whose parties are of relative equal stations. Hence, there remains a need to explore more diverse contexts such as familial, professional, or adversarial communications, particularly those between unequal parties like superiors and subordinates, professors and students, or, in our case, reviewers and authors. Extant work has shown, for instance, that high-power individuals tend to use "I" less often, instead favoring more "we" and "you" usage[52]. In a similar vein, our study enriches our understanding of "you" usage in two-way communications that are both professional and hierarchical.

Moreover, by revealing the link between language and review outcomes, we contribute to the emerging field of science of science, which scientifically probes the practice of science itself[53,54]. Regarding the peer review process, several often-overlapping science of science sub-fields, such as bibliometrics, scientometrics, and metascience, have accumulated important insights into how scientific publication works, what potential biases exist, and how to ensure rigorous, transparent outcomes[55–57]. Through the present work, we underscore that perspectives and methods of language study can bear promising fruit in science of science, and we contribute to the few extant works that have already begun to explore this front (finding, e.g., that scientific papers often use generic, overgeneralized language that signals impact at the cost of precision)[5].

Several limitations in our data should be noted. To begin, the lack of pre-1st-round reviewer comments prevents direct verification of the parallel trend assumption for DID analysis. As a result, the randomness of "you" and non-"you" usage poses a limitation in our data (we have, however, employed such methods to address this issue as PSM, Heckman model, permutation test, and behavioral experiment). Moreover, our dataset comprises only papers eventually published, leading to potential selection biases due to the absence of review reports from rejected submissions or those authors opted not to pursue. Additionally, since publishing review correspondence in *Nature Communications* was optional before November 2022, our data (April 2016 to April 2021) only include authors who opted for publication. These limitations could hinder our ability to analyze "you" usage in, say, more conflictual communications, despite its well-established potential to convey confrontation (e.g., challenging, blaming, or finger-pointing)[2,16,17,58]. Likewise, selection biases in our data also prevent us from comparing "you" usage in accepted versus rejected manuscripts, or between authors who did versus did not choose to publish their review records. Thus, we encourage future research to explore diverse datasets to expand on our findings.

Furthermore, in this study, we interpret the decreased "I" usage by reviewers following authors' "you" usage as indicative of a reduction of self-focused attention. However, we recognize the complexities around this inference[59], as "I" language may also signify language concreteness[1] and self-disclosure[15], contributing to a more personal conversation. While this alternative account is unlikely to contradict our findings due to extensive triangulation, we nevertheless call on future research to delve deeper into first-person usage in written communication.

## Methods

### Ethics
This research is approved by the Office of Research and Knowledge Transfer at Lingnan University and complies with all pertinent ethical regulations.

### Peer review data
We sourced peer review data for all papers from April 2016 to April 2021 directly from *Nature Communications*. Each paper's Supplementary Information section typically hosts its peer review file, which we downloaded using a custom Python (v3.7) script. These files, originally in PDF format, include both reviewer comments and author responses. To create a paper-level peer review dataset, we first separated reviewer comments from author responses for every review round and created separate TXT files for both. We then generated the variables used in our analysis for each paper by review round employing text mining techniques.

To construct the panel data for studying our proposed effects, we employed several automated text analysis techniques to generate desired variables. Specifically, we leveraged Python packages such as TextBlob and NLTK, as well as R packages including sentiments and topicmodels. These methods are well-established in the fields of natural language processing and computer science and are widely adopted in social science studies.

### Sentiments of reviewers' comments
We generated the following four sentiment metrics for reviewer comments. Two of these capture positivity, while the other two capture negativity.

**Python-based positivity.** Positivity is also known as "polarity" in Python and calculated by the TextBlob Python package. TextBlob calculates how positive a reviewer comment is on a scale ranging from −1.0 (highly negative) to 1.0 (highly positive). This calculation is enabled by TextBlob's built-in lexicon, which contains a collection of words and their part-of-speech meanings.

**R-based positivity.** Using the sentimentr package in R, we gained an alternate metric of review positivity, which is also gauged on a −1.0 (highly negative) to 1.0 (highly positive) scale.

**Python-based negativity.** Utilizing Python's NLTK package, we derived the negativity of a review. This approach leverages the VADER (Valence Aware Dictionary and Sentiment Reasoner) sentiment analyzer to evaluate each review's negative emotion scores on a 0 (not negative at all) to 1 (very negative) scale.

**Manually coded negativity.** Following Delgado et al.[60], we incorporated the 30 negative words most frequently employed by our sampled reviewers. We also introduced other negative words that recurrently appeared in our dataset, resulting in a compilation of 92 negative terms. To measure negativity, we determined the occurrence rate of these negative words (scaled by dividing by 100). The scale thus starts at 0 (not negative at all) and increases by 0.01 (or 1%) each time one of the 92 negative words is used.

### Indicators of personal and engaging conversations
The following four variables serve as indicators of a personal and engaging conversations:

**Subjectivity.** Assessed using the TextBlob Python package, again using its built-in lexicon. This measure scales from 0 (very objective) to 1 (very subjective). For illustrative examples of varying subjectivity in reviewer comments, see Supplementary Note 5 and Supplementary Table 10.

**First-person singular pronoun usage.** Quantified by counting occurrences of terms like "I," "me," "my," and "mine" within a reviewer report.

**Word complexity.** Captured by the average number of syllables per word in a peer review report. More syllables per word indicates a more complex vocabulary. For examples of complex and simple words, refer to Supplementary Note 5 and Supplementary Table 11.

**Reviewer engagement.** Deduced from the proportion of the "engagement topic" in a reviewer report. This proportion is obtained by employing the Latent Dirichlet Allocation (LDA) model, a well-established method in natural language processing that uncovers latent topics within a collection of texts.

In our context, the texts in question are the reviewer reports. The LDA model assumes that each report comprises several topics (with the combined probability of all topics being 1) and that every topic is a discrete probability distribution over all words. By implementing the LDA model with a predetermined number of topics, document–topic and topic–word pairs can be formed based on the words included in each reviewer report, allowing us to identify latent topics.

To implement the LDA model, we followed a data preprocessing approach similar to those used in recent studies[61]. Initial steps involved the removal of stop words (e.g., "and," "or"), numbers, and punctuation. We also use stemmed and lower-case words for consistency. We then employed the R package topicmodels to assess the model performance and estimate an appropriate number of topics. Specifically, after experimenting with topic counts ranging from 10 to 100 (at 10-topic intervals), we determined 40 to be the optimal number in that it has the lowest perplexity score. With topic number set to 40, the engagement level was subsequently formulated as:

$$\ln(\% \text{ of engagement topic} \times \text{number of words in the reviewer report}) \tag{1}$$

where *% of engagement topic* is the probability or proportion of the engagement-related topic in the review text calculated by the LDA analysis. *Number of words in the reviewer report* is the total word count in each review text.

## Model

We employ the difference-in-differences (DID) model to identify the impact of second-person pronouns on various outcome variables of interest. Specifically, we estimate the following model:

$$y_{it} = \alpha + \beta_1 \text{Response\_with\_you}_{it} \times \text{After\_response}_{it}$$
$$+ \beta_2 \text{Response\_with\_you}_{it} + \beta_3 \text{After\_response}_{it} + \boldsymbol{\theta} \mathbf{X}_{it} + \delta_i + \varepsilon_{it} \tag{2}$$

Where $y_{it}$ represents the outcome variable of paper $i$ in round $t$. $\text{Response\_with\_you}_{it}$ denotes whether the author(s) of a paper responded with "you", taking the value of 1 if the response includes "you" and 0 otherwise. $\text{After\_response}_{it}$ denotes whether the observation period is after the response, taking the value of 1 if so and 0 otherwise. $\mathbf{X}_{it}$ is a vector of controlling variables of a paper, including (1) the number of pages[62]; (2) the number of references[63]; (3) the title length; (4) the number of authors[64]; (5) H-index of the first author[65]; (6) the gender of the first author[61]; (7) the last initial of the first author[66]; (8) the positivity of authors in the 1st round of review; (9) the friendliness of authors in the 1st round of review; (10) the positivity of reviewers in the 1st round of review; (11) the month the paper was published[67]; (12) the year the paper was published[67]; and (13) the discipline to which the paper belongs (*Nature Communications* identifies five disciplines: biological sciences, physical sciences, health sciences, earth and environmental sciences, and scientific community and society). $\delta_i$ is the paper fixed effects, controlling for the potentially

unobserved paper-level factors. $\varepsilon_{it}$ is a random error term. The coefficient $\beta_1$ is our coefficient of interest, examining the differential effects of responses with and without "you" (on various outcomes) before and after the response. We find that the residuals for DID models approximate a normal distribution, and the variance of the residuals is stable across different levels of the independent variables, as exemplified in Supplementary Fig. 9.

## Behavioral experiment

**Pre-registration.** The behavioral experiment was pre-registered on April 28, 2023 (Pacific Time) with AsPredicted (https://aspredicted.org/9yw2f.pdf). Here, we disclose a total of two deviations from the pre-registration protocol. First, the reported mediation analysis was not originally included in the protocol and was added in response to a review comment. Second, the actual sample size exceeded the pre-registered target by one participant, as explained below.

**Participants.** We recruited 1601 Amazon Mechanical Turk panelists via the CloudResearch platform, who participated in the study for monetary compensation. No statistical method was used to pre-determine sample size. All participants provided informed consent before participating in the study.

The pre-registered target sample size was 1600. However, due to CloudResearch's process for determining sample size, which is outside our control, the study eventually yielded 1601 participants. This deviation was anticipated and noted in our pre-registration.

The participant gender distribution is as follows: Of all 1601 participants, 901 (56.3%) self-identified as female, 676 (42.2%) as male, and 24 (1.5%) as non-binary or chose not to disclose their gender. While we did not plan for a priori gender-based analysis, we have included the results of *post hoc* analyses in Supplementary Method 1, in compliance with the editorial policies of the Nature Portfolio (as of November 12, 2023).

**Data exclusion.** Our pre-registration dictates that data would be excluded from analysis if flagged as fraudulent by Qualtrics, the survey platform used for our study. However, Qualtrics' Expert Review function did not detect any fraud that would warrant data exclusion. Consequently, no data were excluded from the analyses.

**Procedure.** After providing informed consent, all participants were asked to read a brief introduction to the peer review process. The introduction read "Peer review is a process all academics need to go through if they want to get their research work published. When a researcher submits a research paper to an academic journal, the paper is subject to an independent assessment by other field experts called the reviewers (whose role we ask you to play here)."

All participants were then asked to imagine that they had recently reviewed a manuscript for an academic journal. To provide sufficient realism, this hypothetical manuscript was very loosely adapted from a 2020 paper published in *Nature Communications*[68], selected due to its subject matter being easily understandable for laypersons. Specifically, participants were told that "This work examines the possibility that people with more emotional experience (joy, anger, distress, etc.) also have richer emotional vocabulary (i.e., words describing states of emotions) in their language usage."

All participants were then instructed to imagine that after reviewing the manuscript, they wrote the following comments to the author of the paper:

- Overall, the paper presents an interesting theory and is well-written.
- The studies included in the paper are well designed and the interpretation of data is generally convincing.
- That being said, detailed criteria on what counts as "emotional vocabulary" are lacking. For instance, the usage of such words as

"alone" or "bad" does not necessarily carry emotional connotations. As a result, the inclusion of such words in data analysis may prove problematic.

- The contribution of the work is insufficiently elaborated. To this end, the paper needs to better explain why this work helps advance what the field already knows.

Note that no "you" language was presented in these comments.

All participants were then informed that they had now received the author's responses. The responses were otherwise identical, save for how the participants (i.e., the reviewers) were addressed. By this design, participants were randomly assigned to one of the two conditions (i.e., "you" and non-"you"). Specifically, participants in the "you" [non-"you"] condition read:

We appreciate your [the reviewer's] comments, which we find very useful. With regard to the questions you [the reviewer] raised:

- You [The reviewer] advised us to provide details on how emotional vocabulary is determined. Building on your [the reviewer's] advice, we now include a thorough discussion of your [the reviewer's] concern over this issue, and lay out the selection procedure of those words in the manuscript.
- In this discussion, we also address your [the reviewer's] concern that some words are not applied solely to emotional experience.
- You [The reviewer] suggested that the contribution of this work be differentiated from existing research. Following your [the reviewer's] suggestion, we explain how this work advances the understanding of emotions and affective language.
- As per your [the reviewer's] recommendation, in this revision we also further elaborate the contribution of this work in the discussion section.

The participants were unaware of their assigned condition and were not cognizant of the existence of the alternate condition to which they were not assigned. The investigators, on the other hand, were not blinded to allocation during experiments and outcome assessment.

Participants were then prompted to evaluate the how personal and engaging they found the conversation to be on a 4-item, 7-point Likert scale (1 = strongly disagree; 7 = strongly agree; Cronbach's $\alpha = 0.86$): "In general, I find the conversation between the parties engaging," "The author is engaging in a personal conversation with me," "The correspondence between the reviewer and the author feels conversational," and "I find the author personable." Participants also rated the positivity of the author's response on a single-item Likert scale "My overall impression of the author's response is positive."

## Reporting summary

Further information on research design is available in the Nature Portfolio Reporting Summary linked to this article.

## Data availability

All data necessary for reproducing the results presented in this paper have been deposited in OSF (https://doi.org/10.17605/OSF.IO/XWYS4)[69].

## Code availability

All code necessary to reproduce our analyses are available at OSF (https://doi.org/10.17605/OSF.IO/XWYS4)[69].

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

## Acknowledgements

C.C. is supported by the Research Grants Council of Hong Kong (13501722) and the Lam Woo Research Fund (F871223) at Lingnan University. Y.L. is supported by the Research Grants Council of Hong Kong (13503323), the Lam Woo Research Fund (LWP20020) and Faculty Research Grant (DB23A5) at Lingnan University, and the National Natural Science Foundation of China (72271060). C.M. is supported by the General Project of National Natural Science Foundation of China (72074045).

## Author contributions

C.M., Y.L., Z.S. and C.C. designed research; Z.S., C.C., Y.L. and C.M. performed research; Z.S., C.C. and S.L. collected and analyzed data; and Z.S., C.C. and Y.L. wrote the paper. All authors wrote, edited, and revised the manuscript.

## Competing interests

The authors declare no competing interests.
