## [Peer Review File · Nature Communications]

Behavioral consequences of second-person pronouns in written communications between authors and reviewers of scientific papersREVIEWER COMMENTS

Reviewer #1 (Remarks to the Author):

This article presents an intriguing set of results from a publicly available dataset containing roughly 25,679 peer review exchanges for the journal Nature Communications. The paper is well-written and easy to read. The authors employ a difference-in-differences approach to test how an author team's use of "you" in their initial response to reviewers affects reviewers' subsequent correspondence. Overall, the authors find that using "you" is associated with positive outcomes, including responses from reviewers that are more positive and less negative; show more engagement and subjectivity; and are shorter overall and include fewer questions. The authors also rule out the possibility that responses from reviewers are more self-focused, illustrating that first-person singular pronoun use goes down in reviewers' response. Finally, the authors also show some of these effects are strengthened when the reviewers' *first* correspondence to the authors contains "you."

This is a provocative paper that has implications for how people in academia might leverage language to help them along the way in the peer review process. It also advances our understanding of the science of science (see also DeJesus et al., 2019 for an interesting paper on how the language that authors use in their paper titles affects the perceived importance of the results).

I have several comments and questions that the authors may wish to consider as they continue this work.

Big Picture

Although the authors link their findings to the broader literature on how small linguistic choices can carry interpersonal implications in the discussion section, I believe they could spend more time discussing possible implications of these findings and how they might generalize to other interpersonal dynamics outside of a peer review process, which is fairly specialized.

A huge strength of this paper is that it studies the interpersonal effects of "you" by utilizing naturally occurring written conversations. The authors control for a wide variety of control variables, but there is still the reality that enormous variability likely exists in the texts that the authors analyzed. In opening the paper, the authors provide an example wherein the same sentiment is expressed, either using "you" or not ("the issue you brought up..." vs. "the issue the reviewer brought up"). This example begs the question of direct causality – that is, did the authors consider designing a causal experiment where they hold the content of a statement constant but vary whether the second person address is included vs. a third person address? This would help isolate the causal role of "you." Follow-up experiments could also get at the underlying mechanisms driving these effects, providing more direct evidence into whether the reviewer feels a heightened connection to or sense of obligation to the author when being addressed with "you," which alters the way in which they are engaging with the work. Or, whether, as the authors posit, "you" increases engagement. Additional data exploring these issues would strengthen the paper.

In this vein, the authors say that "you" usage leads to outcomes such as fewer questions etc. because it leads to a more engaging conversation. However, I did not see this mediational pathway directly tested.

As a related point, I did not see the authors include other control variables that might covary with the authors' use of "you" in their analyses. For example, are authors' responses with "you" also more friendly, positive, less confrontational etc.? The authors could consider adding some such additional control variables that get at these considerations.

Methods and Additional Analyses Questions

Related to the point above, do author features predict use of you? For example, gender, rank? I

understand that this is not the main focus of the paper, but I am curious and it seems relevant to understanding how these interpersonal dynamics play out, particularly in a context (i.e. the peer review process within academia) with power structures and imbalances.

I was wondering whether there are features of the *initial reviews* that predicted "you" usage vs. not in the authors' responses? That is, were the authors responding to a tone that was already set by reviewers? The authors actually address this partially with the differences-in-differences-in-differences analysis, where they find that some of the effects are strengthened when the reviewers initially used "you" but I think this could be approached in more depth. I am also wondering about "matching." The authors emphasize the interactive conversational dynamics in their title – can they do more to explore this? For example, is it the case that matching "you" usage (i.e., reviews include "you" \diamond response to reviewers includes "you" (vs. response to reviewers does not include "you")) is particularly beneficial? If the DDD approach gets at this, please let me know.

I was also curious whether authors used "you" to address all reviewers or just some? For example, might they have used "you" only to address reviewers who they intuited were more receptive to the work?

Can the authors look at any other behavioral outcome variables, like the number of back and forths between the authors and the review team before eventual acceptance? It would be interesting to look at acceptance vs. rejection as a function of "you" use, but rejected papers are not available, presumably.

Minor Comments

Was the study pre-registered? If not, can the authors state this.

Can the authors provide an indicator of effect size?

The authors provide descriptives on the proportion of author responses that contained you (38%) vs. did not in the Supplement. I think it's worth including this in the main text.

The authors say this is the first paper to look at the effect of "you" in a bilateral conversational setting. I believe there has been a decent amount of work looking at the effects of pronouns (I, we, you) in dyadic conversations between romantic partners, particularly in the context of coping etc. The authors may want to qualify or peel back their claim.

Overall, this is an exciting study and I wish *you* good luck as *you* continue this work :)

Reviewer #2 (Remarks to the Author):

In this manuscript, the authors studied the effect of using "you" in written communications (i.e., referee report and response of Nature Communications) on the number of questions, number of words, positive/negative embedded in the report. Furthermore, the authors find that using "you" in the response seems to lead a more personal conversation, as it increases subjectivity, and reviewer engagement, but reduces word complexity and first-person singular pronouns usages. Finally, the authors also find that if "you" was initiated by reviewers, the engagement and subjectivity seem significantly stronger. I read the manuscript with great interest. The topic is really interesting, and it contributes to the field of Science of Science and scientists' behaviors of certain language use. The manuscript is well written. Overall, I believe this manuscript deserves publication after addressing my comments/suggestions below.

Major points:

1) The authors divide referee reports according to the usage of "you", and it seems a binary separation. I am wondering whether a report with MANY "you" shows even less questions/words and

more positive comments.

2) The manuscript is full of causality words, e.g., lead to. I suggest the authors use these words with caution, as DID doesn't speak to causality and doesn't untangle endogenous issues. Moreover, understanding the underlying mechanisms why such effects exist is necessary. If controlling for subjectivity/work complexity/reviewer engagement, does the main effect shrink?

3) Throughout the manuscript, there are many measurements that are not crystal clear to me, such as word complexity/subjective. I believe some qualitative evidence is necessary. For example, the authors could provide some examples with more subjective sense as well as less subjective sense. OR, some complex words as well as some simple words.

4) The authors used LDA to identify topics in order to measure review engagement, and find that reports with "you" seems to show more reviewer engagement. However, it is not clear to me how to measure the engagement, is it the probability to have topic 11? Also, it is great if the authors can provide some test by replacing the relevant topic to unrelated topics.

5) For table 4, the authors mentioned "most of the effects become significantly stronger". I didn't see empirical evidence, and the authors should elaborate more on this.

Minor points:

1) I suggest to put the description of control variables in the main text, as it is important to know what is the control variable.

2) In the Tables, it is good to put the coef. of Response with "you", and After Response as well.

3) If necessary, the authors should also interpret the coef. For example, reports with "you" show XX less words and XX less questions.

Reviewer #3 (Remarks to the Author):

This is a fascinating and quite promising project. The underlying idea is powerful and is something many authors and reviewers have thought about (including me). The sample is remarkable and the methods are generally strong. Despite its considerable strengths, the results are difficult to interpret. To the degree the authors would like to reach a broad audience, I would encourage them (that is, you) to reframe the results with social scientists in mind.

Your central argument is that the use of "you" sets up a more personal relationship between the author and reviewer. Until the last paragraph of the results section (lines 285-297), the paper is going on the assumption that the relationship is essentially built by the author in response to the first set of reviews. But if this is a true interaction, isn't it actually set up initially by the first reviews themselves? If I, as a reviewer, talk to you (the author) in a personal way, hasn't the dialog begun? This should not be an afterthought in the paper. It should be part of the cascading interaction of the study: The initial reviews affect the authors' first responses which, in turn, affect the second round of reviewers' comments which then affect the authors' second responses, etc. (In theory, you could start with the original paper but, in reality, virtually no "you" words will be in most papers.)

The analytic approach should more closely follow/reflect the social processes you are testing. Using the cascade idea, what are the mean word count (WC) and question count (QC) of authors who are responding to reviews with and without you-words? And then, what are the mean WC and QC of reviewers who respond to authors who do and who do not use you-words? One way to approach the analyses might be something like this:

Level 1: Reviews that Use You-Words Reviews that Don't Use You Words

Level 2: Authors do ... A's don't use ... Authors who do vs don't use You

Level 3: Rev Do/Don't Revs Do/Don't ... Revs Do/Don't Revs Do/Don't

It would be nice to see the actual means either in the text or supplemental information. Adjusted means with the multiple covariates could be used as well. Ultimately, we need to get a sense of the numbers. How big are these effects? The R2 you report undoubtedly include all the covariates. We need to see the specific effects of you-words. Small effect sizes are likely to result but given the question and your use of such glorious real world data, that's OK. Note also that there is an entire industry that is devoted to conversational data that deals with recursive interactions that attempt to disentangle the two sides of a natural conversation (see the important work of DA Kenny, e.g., 2020).

A related question concerns the reliability of you-word usage among multiple reviewers of the same paper. Assume a paper has three reviewers, how frequently do reviewers 1 vs 2 vs 3 use you-words? Similarly, in the authors' responses to the reviews, do the authors selectively respond to those reviewers who use you-words? What these questions get at concerns the nature of the person who uses you-words. Is there a certain type of personality that tends to use you-words or is this primarily situational? Does a certain type of author or paper provoke you-word use?

Secondary comments

Perhaps the terms of "difference-in-difference" or "DID" and, later, DDD, are common in the authors' professional specialization. They are not intuitive to an outsider. Why are they called DID? Difference from what? Surely there is a better way to label the phenomenon you are talking about.

The LDA analyses are a nice touch. It's not clear how much they contribute to the main point of the paper however.

Line 123. First graphic should include Ns

The tables are difficult to interpret. It's not clear what columns 1 and 2 represent. I initially thought they referred to "No You Use" and "You Use". But no. I would like to see the actual mean usage of you-words, either in the main text or supplemental materials.

The courtesy "you" words are quite interesting. Again, it would be helpful to include the means. In other words, what percentage of all you-words are accounted for by courtesy words? To what degree do courtesy you-words correlate with non-courtesy you-words? If they are correlated at the manuscript level at, say, .30 or higher, it could be argued that they are reflecting the same or similar process (e.g., awareness of the others' perspective as opposed to pure courtesy if the correlation is below .30).

In summary, this is a project that has the potential to make a big splash both theoretically and practically. By changing the analytic approach, it will have a far greater impact on researchers in the social sciences.

Jamie Pennebaker

Reviewer #4 (Remarks to the Author):

Comments for the author(s):

This paper is, I believe, interesting and important. I am not aware of any other contributions that cover this same kind of area (at least not in the way that this one does), and so it is a novel and significant contribution, I believe. The paper reports noteworthy results focusing on the interactions between authors and reviewers, and in particular the language used in addressing each other with regard to use of the second person, "you". The findings demonstrate that use of "you" and similar

constructions lead to more a positive, personal and engaging conversation between the reviewer and authors.

Your paper is clearly written and does a good job of presenting your data with clarity. I believe that reviewers with particular expertise in the methods you have used will also be commenting on the methodological aspects of the paper in detail, whereas my remarks relate to more general issues, so as not to overreach my expertise. Having said that, the study seems to be well conducted and, as far as I can tell, robust. The paper provides a clear account of the methods used which appear to be sound. The inferences drawn from the data also seem reasonable, although I do have some minor suggestions for how the paper might be improved.

I would like to see the review of literature expanded a little, since at the moment it is very brief. This would help provide more context for the study. At present, the second paragraph of the introduction (lines 41-51) contains reference to a number of previous studies (citations 1-7) but there is very little commentary on them and the way in which your study relates to them is only described in a very elliptical way. There is also some reference to previous studies in lines 72-90, although these are presented more in relation to the way you have set up your own study rather than a commentary on the literature as such. The Discussion mentions connections with fields of bibliometrics, scientometrics and metascience, but I would suggest that it would be useful to see these explored upfront to a greater extent.

I would also like to see you define more clearly what you mean (and don't mean) by "subjectivity". In the paper, you use the term in a particular sense to mean more personally engaged language (lines 141-145 indicate that). However, 'subjectivity' can be used in a broader sense, often associated with the idea of personal bias. The Cambridge University online dictionary defines "subjectivity" as "the influence of personal beliefs or feelings, rather than facts". You are not using the term in this broader sense, but it would be useful for you to define your usage more explicitly, and to exclude other connotations.

I have a number of comments which relate to reviewer use of "you", as opposed to author use. The focus of your analysis is on author use. Your coverage of reviewers' use of "you" is rather incidental, although you do make some interesting remarks about it from line 275 onwards. However, at other points in your analysis it does feel rather one-sided. I would like to see more about reciprocity between author and reviewer in terms of language, built into the paper more consistently, if your data allows this.

Your comments on "positivity of review" (from line 170 onwards) raise the question of the origin of "you" language in the review correspondence. You mention author use of "you" as resulting in more positive reviews, but authors are, of course, responding to reviewer comments at that point. You augment this analysis later, from line 275 onwards, with comments about reviewer initiation of "you" language. This does address some of the questions that arise from the earlier analysis. However, I believe the paper would benefit from more analysis of different aspects of author-reviewer reciprocity, where it can be built in. In your Discussion you mention authors following suit in language in responding to reviewers, once again this feels rather one-sided – how does this work the other way round? What are the characteristics of usage that tend to be reproduced in the process of one actor following the suit of the other?

I would be interested to see more detail on the comment that, "...in our data, authors' usage of "you" is in fact also associated with reviewers' decreased usage of first-person singular pronouns (i.e., "I", "me", "my")". My question again is how does this relate to reviewer use of "you"? You go on to comment on reviewer use of "you" later. Where there is reciprocal use of "you" between authors and reviewers, I would expect use of the first person, "I", to naturally accompany this. As language becomes more personal, the use of the first person might be thought to increase along with the second person. However, this does not seem to be the case in your data and so it would be useful to

see this discussed.

A final point related to this is that of confrontational use of "you" and "I". Sometimes in confrontational situations the use of "you" versus "I" can be seen as heightening tension (see Rogers, S. L., Howieson, J., & Neame, C. (2018). I understand you feel that way, but I feel this way: The benefits of I-language and communicating perspective during conflict. *PeerJ*, 6, e4831. <https://doi.org/10.7717/peerj.4831>). Do you see any evidence of this or have any comments on it in relation to your findings?

I would be very interested to see examples of sentences use the illustrate your key points, even if in appendices. As it is, the focus of reporting is on individual words rather than on sentences, and whilst I realise this is a consequence of the way in which you have conducted your analysis, it would be useful to have some example sentences included.

The supplementary material is helpful and clear, although I am not sure of the value of the word cloud which, if it comes from all of your data, doesn't really tell us much.

The writing of the article is clear and it is very well presented. There are a few minor proofing areas that need correcting.

Reviewer #5 (Remarks to the Author):

Comments are included in the file attached.

Thank you for the opportunity to review “Two can play at that game of ‘you’: The behavioral consequences of using second-person pronouns in written communications”. This paper uses peer review correspondence collected from *Nature Communications* to examine the relationship between authors’ use of second-person pronouns in their initial response to reviewers and various characteristics of reviewer responses in subsequent rounds of peer review. In particular, the authors utilize a differences-in-differences (DID) approach in attempt to identify the causal effect of second-person pronoun usage on the structure, sentiment, style of reviewer responses. While the paper tackles an interesting question about the nature of author-reviewer interactions in the peer review process, a key weakness of the paper lies in its statistical methodology, which is, in many places, weak and/or inadequately explained. For that reason, I believe the manuscript would benefit from a major revision.

Some general feedback as well as several specific comments are included below.

General

- Throughout the paper, the authors utilize a DID approach to examine the effect of “you” versus non-“you” usage on aspects of reviewer correspondence. While I agree that this is an appropriate model for the problem at hand, the authors fail to acknowledge any of the (strong) assumptions required for inference in this setting and offer no discussion of the methodological limitations of their analysis. Without addressing these critical issues, I am skeptical about the credibility of the paper’s claims, particularly those regarding causality.
- Related to the issue above, the authors define the intervention (here, the use of 2nd person pronouns in an authors’ first round of responses to reviewer comments) in a manner that allows units to self-select into the treatment and control groups. As a result, there is tremendous potential for confounding due to pre-existing differences between groups. For instance, perhaps authors who receive generally positive feedback in their first round of reviewer comments are more likely to use “you” in their responses compared to those who receive more critical feedback. Similarly, it seems that an authors overall writing style or the strength of their arguments could explain the relationship between pronoun usage and aspects of reviewer responses. While I don’t believe these issues are necessarily insurmountable, they do enough of a philosophical threat in the present analysis to warrant consideration.
- Much of the Discussion section presents background information that connects the present analysis to existing literature. As a result, this section reads more like a second Introduction than a typical discussion. I would recommend restructuring this part of the paper to include a more detailed examination of the paper’s main findings as well as a discussion of the paper’s strengths and weaknesses, with a particular emphasis on the methodological limitations.

Specific

- On line 112 the authors state that their data includes “13,359 published papers that account for a total of 29,144 round of review”. However, in Table S1, the authors present summary statistics for a sample of $N = 26,679$ documents and in the majority of their analyses they list a sample size of $N = 26,640$. Where are these discrepancies coming from?
- Related to the above, did all papers included in the analysis make it through at least 2 rounds of review before acceptance? If not would be helpful to present the total number of papers represented in each phase of the review process.
- On lines 118-122, the authors state that they focus on the first review-response-review process for their analysis. However, Fig. 1 shows multiple rounds of review. It is unclear to me whether the text comments from rounds 2 and after used for any part of this analysis. If not, the authors might consider updating or removing Fig. 1 to avoid confusion.
- In Table 1, what is the purpose of including the “basic DID regression” without controls or fixed effects for these particular outcomes?

- On line 251, when the authors state that they employ “LDA on our corpus to identify topics relevant to the reviewers’ engagement”, I am confused about what corpus they are referring to. Was LDA run on only the first round of reviewer comments? Only the second round? Some combination?
- For the LDA analysis beginning on line 250:
 - I assume the topic model is estimated using reviewer comments from both the treatment and control group. Since: 1) there are considerably more observations in control and 2) reviewer comments tend to be longer in the control group, the estimated topic model is likely to be dominated by language that may be specific to the control group. A structural topic model might be more appropriate to address this issue.
 - It would be helpful to show the distribution of document-level topic proportions within the chosen topic. Is prevalence within this topic generally high or low, and with how much variability?
 - The authors seem to make a very strong claim that: 1) their topic model is stable, and 2) the estimated topic proportions for their identified topic are an accurate reflection of reviewer engagement. Is there any precedent in the literature to support this? Why not test for differential use of a subset of “high-engagement” words (e.g., based on word counts), which would avoid the complications described above?
- On line 290, in what sense do the effects become “significantly stronger”? Is this statement based on the magnitude of the estimated effects, overall model fit, or something else?
- Equation (2) should include a plus sign at the end of line 421

Point-by-point response to reviewer comments

Reviewer #1 (Remarks to the Author):

*This article presents an intriguing set of results from a publicly available dataset containing roughly 25,679 peer review exchanges for the journal Nature Communications. The paper is well-written and easy to read. The authors employ a difference-in-differences approach to test how an author team's use of "you" in their initial response to reviewers affects reviewers' subsequent correspondence. Overall, the authors find that using "you" is associated with positive outcomes, including responses from reviewers that are more positive and less negative; show more engagement and subjectivity; and are shorter overall and include fewer questions. The authors also rule out the possibility that responses from reviewers are more self-focused, illustrating that first-person singular pronoun use goes down in reviewers' response. Finally, the authors also show some of these effects are strengthened when the reviewers' *first* correspondence to the authors contains "you."*

This is a provocative paper that has implications for how people in academia might leverage language to help them along the way in the peer review process. It also advances our understanding of the science of science (see also DeJesus et al., 2019 for an interesting paper on how the language that authors use in their paper titles affects the perceived importance of the results).

I have several comments and questions that the authors may wish to consider as they continue this work.

We are heartened to learn that you find that our paper is provocative, has important implications, and contributes to advancing the science of science. We are most grateful for your thoughtful and penetrating comments. Furthermore, we appreciate your introduction of an interesting paper, DeJesus et al. (2019). Indeed, this important work exemplifies how language usage influences the evolution of science, and we have now included this important work in the Discussion (line 447; reference 60). In the sections that follow, your comments are presented in italics, each followed by our response. We hope you find that our revision meets your expectations.

Big Picture

Although the authors link their findings to the broader literature on how small linguistic choices can carry interpersonal implications in the discussion section, I believe they could spend more time discussing possible implications of these findings and how they might generalize to other interpersonal dynamics outside of a peer review process, which is fairly specialized.

Thank you for your valuable suggestion, which has guided us to better elaborate on the implications of our findings. Following your comments, we have now expanded our discussion. In a nutshell, beyond the direct and specific implications for academics, we have also discussed how our findings may be relevant to other (perhaps formal) written communication scenarios, such as business or political correspondence (lines 406-418). Relatedly, your suggestion also helps us better situate our

findings in the extant literature, extending the research on bilateral “you” usage beyond its current focus on intimate relationships (lines 426-437).

A huge strength of this paper is that it studies the interpersonal effects of “you” by utilizing naturally occurring written conversations. The authors control for a wide variety of control variables, but there is still the reality that enormous variability likely exists in the texts that the authors analyzed. In opening the paper, the authors provide an example wherein the same sentiment is expressed, either using “you” or not (“the issue you brought up...” vs. “the issue the reviewer brought up”). This example begs the question of direct causality – that is, did the authors consider designing a causal experiment where they hold the content of a statement constant but vary whether the second person address is included vs. a third person address? This would help isolate the causal role of “you.” Follow-up experiments could also get at the underlying mechanisms driving these effects, providing more direct evidence into whether the reviewer feels a heightened connection to or sense of obligation to the author when being addressed with “you,” which alters the way in which they are engaging with the work. Or, whether, as the authors posit, “you” increases engagement. Additional data exploring these issues would strengthen the paper.

In this vein, the authors say that “you” usage leads to outcomes such as fewer questions etc. because it leads to a more engaging conversation. However, I did not see this mediational pathway directly tested.

Thank you for encouraging us to conduct a behavioral experiment to buttress our causal claims and provide more insights into the underlying mechanism. Closely following your suggestion, we conducted a behavioral experiment (N = 1,601). Specifically, participants who assumed the role of reviewers were asked to read otherwise identical author responses, varying only in the usage of second- versus third-person pronouns. For instance, participants read “As per your recommendation” in the “you” condition, and “As per the reviewer’s recommendation” in the non-“you” condition. We discovered that *ceteris paribus*, participants find responses using (vs. not using) second-person pronouns to be more positive, and their conversation with the author more personal and engaging (for details see lines 335-361). This is congruent with the conclusions drawn from our other analyses leveraging the large-scale secondary dataset.

We also entirely agree with you that a controlled experiment allows us to directly test our proposed process. Consistent with our proposition, participants in the “you” (vs. non-“you”) condition viewed their communication with the author as more engaging and personal. Subsequent mediation analysis further demonstrates that the positive relationship between “you” usage and the positivity of review outcome is mediated by the extent to which participants deem their communication with the author engaging and personal. Moreover, heeding your advice, we have also investigated the effect of “you” usage on positivity can be attributed to (1) a heightened connection between the participant (i.e., the reviewer) and the author, (2) participants’ increased sense of duty towards the author, or (3) a confrontational undertone of second-person pronouns (a perspective suggested by Reviewer 4). Our analyses yield no substantial support for these accounts (details can be found in SI J).

Taken together, the inclusion of the behavioral experiment not only triangulates our existing analysis on “you” usage but also furnishes direct causal support for the effects and mechanisms detailed in this work. Your invaluable feedback has thus substantially enhanced this work’s robustness and internal validity.

As a related point, I did not see the authors include other control variables that might covary with the authors’ use of “you” in their analyses. For example, are authors’ responses with “you” also more friendly, positive, less confrontational etc.? The authors could consider adding some such additional control variables that get at these considerations.

We appreciate your suggestions and agree that it is important to include additional controls. Following your advice, across all our models, we have now controlled for how friendly authors’ responses are (as captured by the frequency of a hand-coded collection of friendly words within the authors’ feedback), as well as the positivity of author responses (i.e., the positive sentiment measured by Python package *Textblob*). Our observed effects remain stable after controlling for these factors, bolstering our confidence in the robustness of our findings. Please also note that our experimental data (lines 359-361; SI J) indicate that our findings persist when the level of contention is held constant. We hope that our revision has adequately addressed your comments.

Methods and Additional Analyses Questions

Related to the point above, do author features predict use of you? For example, gender, rank? I understand that this is not the main focus of the paper, but I am curious and it seems relevant to understanding how these interpersonal dynamics play out, particularly in a context (i.e. the peer review process within academia) with power structures and imbalances.

Thank you so much for sharing your valuable insights. As per your advice, we delved into potential precursors to authors’ use of “you.” Specifically, we examined if the gender and rank of authors predict their propensity to use “you.” For gender determination, we probabilistically inferred the gender of authors from their names, utilizing the Social Security Administration (SSA) database (Si et al. 2023; Su et al. 2023; Sun et al. 2023). As for rank, as it is impractical to manually retrieve precise professorship details of more than ten thousand authors in our dataset, we approximate author rank using the H-index (a citation-centric metric denoting scientific impact) supplemented from the Web of Science database.

We then estimated the influence of these author features on “you” usage. As shown in Table D-6 (of SI D), we did not find a significant gender difference in authors’ “you” usage. Interestingly, the data do suggest that higher-ranking authors (as defined by authors with higher H-indices) are more likely to use “you.” From an applied perspective, this result suggests that seasoned authors, with an extensive publication history, might instinctively recognize the efficacy of employing “you” in fostering positive and engaging dialogues, and thus opt for more frequent use of second-person pronouns during the review process.

This intriguing observation further prompted us to examine the possibility of self-selection bias, wherein certain authors, especially experienced, senior ones, might proactively choose to use “you” in

their responses. To address this potential bias, we conducted a supplementary robustness check employing the Heckman two-stage model (Heckman 1979; Leung et al. 2023). In brief, the first-stage model predicts authors' "you" usage based on covariates such as author features (e.g., gender and rank) and the "you" usage in initial reviews (the specifics of this variable will be elaborated upon in our response to your next question, which is related to the current one). This model yields the Inverse Mills Ratio (IMR), which represents the unobserved determinants of authors' "you" usage. The second-stage main model then formally estimates the effects of authors' "you" usage, correcting for self-selection by controlling for the IMR from the first stage (refer to lines 155-171 of SI D; SI D: Supplementary Table D-6 for further details). The estimated effects from this analysis remain consistent and robust, reinforcing our confidence in the findings. We believe these in-depth discussions present a clearer logic and make greater practical contributions. Again, we appreciate your instrumental comments.

*I was wondering whether there are features of the *initial reviews* that predicted "you" usage vs. not in the authors' responses? That is, were the authors responding to a tone that was already set by reviewers? The authors actually address this partially with the differences-in-differences-in-differences analysis, where they find that some of the effects are strengthened when the reviewers initially used "you" but I think this could be approached in more depth. I am also wondering about "matching." The authors emphasize the interactive conversational dynamics in their title – can they do more to explore this? For example, is it the case that matching "you" usage (i.e., reviews include "you" response to reviewers includes "you" (vs. response to reviewers does not include "you")) is particularly beneficial? If the DDD approach gets at this, please let me know.*

Thank you for encouraging us to look at the relationship between initial reviews and authors' "you" usage in more depth. We also apologize for not having elaborated on the role of DDD in the initial reviews sufficiently. Yes – the DDD analysis have shown that *mutual* "you" usage (i.e., both in "initial reviews" and author responses) is particularly beneficial. That is, engagement derived from authors' "you" usage is significantly higher when the initial reviews also employ "you" as opposed to when they don't. To better explain this effect, in the revised manuscript we have now divided the DDD analysis into two DID. These DIDs estimate the effects of authors' "you" usage in two subsamples: initial reviews that used "you" and those that did not (refer to lines 369-375; SI K: Supplementary Table K-1). As anticipated, the effect sizes of authors' "you" usage are generally larger in the subsample where initial reviews used "you" (e.g., $\beta_{\text{engagement}} = 0.1417$), relative to the subsample that did not (e.g., $\beta_{\text{engagement}} = 0.0613$). This result produces evidence that mutual "you" usage between the reviewer and the author creates a more personal and engaging conversation.

At the same time, it is worth remembering that reviewers' initial "you" usage is *not* necessary for authors' "you" usage to be impactful. Indeed, authors' "you" usage is sufficient to render a conversation more engaging and personal ($p_{\text{engagement}} < .05$), even when initial reviews are devoid of "you." This observation is further corroborated by our behavioral experiment: participants addressed as "you" (vs. not "you") find their conversation with the author more personal and engaging, without addressing the author with "you" in the first place.

On the other hand, we also agree with your observation that “you” in the initial review can at least partially predict authors’ subsequent “you” usage in real peer review contexts. Motivated by your insightful observation, we found a significant positive correlation between the “you” usage in initial reviews and that in author responses. Therefore, for completeness of analysis, we have further included the reviewers’ initial “you” usage as an additional covariate in the first-stage model (predicting authors’ “you” usage), alongside other covariates such as author gender and rank (as we discussed in our response to your previous question). We believe these analyses can generate fruitful insights for readers interested in potential antecedents of authors’ “you” usage (for details see SI D: Supplementary Table D-6).

I was also curious whether authors used “you” to address all reviewers or just some? For example, might they have used “you” only to address reviewers who they intuited were more receptive to the work?

Thank you for bringing up this intriguing perspective. To explore whether authors might use “you” to address only some reviewers, we examined the review correspondences of 1,000 randomly selected papers from the full dataset. We did this because it would be impractical to work with the entire dataset, as it is very challenging to algorithmically locate identifiers that distinguish individual reviewers within the full review file. Of these 1,000 papers, 347 had authors who used “you” in the review process. We then manually identified each reviewer and author(s) for these 347 papers, cataloging their respective “you” usages throughout the correspondences.

For each of these 347 papers, we computed P_{Author} , which is the **proportion** of reviewers the author(s) addressed using “you.” For example, $P_{\text{Author}} = 1/3$ means the author(s) addressed 1 out of 3 reviewers using “you.” This distribution is presented in Panel (A) of Exhibit 1. Similarly, we calculated P_{Reviewer} , denoting the proportion of reviewers addressing the author(s) using “you.” For instance, $P_{\text{Reviewer}} = 1/3$ means that 1 out of 3 of the paper’s reviewers used “you” in their comments. Panel (B) of Exhibit 1 depicts the distribution of P_{Reviewer} .

Particularly notable are the two ends of both figures. To begin, since this dataset includes only those authors who use “you,” there are no instances where $P_{\text{Author}} = 0$ (i.e., authors did not address any reviewers using “you”; the leftmost bar in Panel (A)). In contrast, the majority of papers in this dataset (almost 60%) have a $P_{\text{Reviewer}} = 0$, indicating that for most papers, no reviewer addressed the authors using “you,” despite the fact that at least one of the reviewers had been addressed with “you” (the leftmost bar in Panel (B)). This divergence indicates that a significant number of reviewers do not reciprocate the “you” usage with authors. This non-matching “you” usage is further corroborated by the other end of the figures. Indeed, a significant number of papers (over 40%) address all reviewers using “you” ($P_{\text{Author}} = 1$; the rightmost bar in Panel (A)). This indiscriminate usage of “you” cannot result from authors mirroring a specific reviewer’s “you” usage, as very few papers had all their reviewers use “you” in their comments ($P_{\text{Reviewer}} = 1$; the rightmost bar in Panel (B)).

In fact, out of the 347 papers, only about 10% (34) showed an “exact match” in “you” usage – that is, authors used “you” to address only the reviewer(s) who had initially used “you.” Taken together, this

sample indicates that while exact “you” matching exists, it is rare at the individual reviewer level (as opposed to the overall paper level).

Exhibit 1 The Distribution of P_{Author} and P_{Reviewer}

Note: x-axis not to scale. Though most papers have 3 reviewers (leading to proportions like 1/3 and 2/3), we also observed proportions like 1/5, 1/4, 1/2, or 3/4, as some papers had 4 or even 5 reviewers.

Can the authors look at any other behavioral outcome variables, like the number of back and forths between the authors and the review team before eventual acceptance? It would be interesting to look at acceptance vs. rejection as a function of “you” use, but rejected papers are not available, presumably.

Thank you for the comments. Thanks to the professionalism of the editors and reviewers, the peer review process at *Nature Communications* is generally very efficient: In our dataset, most papers (75.1%) are accepted within two rounds, while a smaller fraction (24.9%) of papers proceed to a third round. Only a minuscule fraction of papers undergo a fourth (0.4%) or fifth round (0.3%). Given the limited variations in review rounds, the effects of “you” usage on them may not be evident in our data. Actually, when we checked whether the number of revision rounds is associated with the “you” usage, the results suggest insignificant relationships (as shown in Exhibit 2 below).

Exhibit 2. The Effect of the “You” Usage on the Number of Revisions

	Number of Revisions	
	(1) Poisson regression	(2) Cox Model
Response with “You”	0.0019 (0.0012)	-0.0021 (0.0020)
Control Variables	Yes	Yes
Observations	13,359	13,359

Note: The DID design is not feasible because only the count (rather than the repeated measure) of the outcome variable is available. The Poisson model and Cox proportional hazards model are applied instead.

Nonetheless, we urge that one interpret these results with caution due to the aforementioned data constraints. In fact, we agree with your hypothesis suggesting that usage of “you” could influence paper rejection rates and review round numbers. Yet, to comprehensively study these aspects, we would require data on paper rejections and perhaps a journal with more variation in revision rounds. We deeply appreciate your point to this interesting angle, which can be potential future research for us.

Minor Comments

Was the study pre-registered? If not, can the authors state this.

We have now updated the preregistration status in compliance with *Nature Communications*’ guidelines. Thank you for pointing it out.

Can the authors provide an indicator of effect size?

Thank you for your excellent suggestion. We have now interpreted the coefficients to meaningfully indicate the effect size where appropriate. For example, reviewers addressed by “you” (vs. non-“you”) raised 3.34 fewer questions and wrote 135.59 fewer words on average (lines 194-197). We have also reported effect sizes for our experiment (lines 335-361). We hope these additions can help our readers better interpret our effects.

The authors provide descriptives on the proportion of author responses that contained you (38%) vs. did not in the Supplement. I think it’s worth including this in the main text.

Thank you for the helpful comment. In our revised manuscript, we detail the proportion of author responses using “you” (37.74%) versus those not using “you” (62.26%). We presented this data in the “Data and Design” subsection of the “Results” section, where we define the treatment and control groups for the DID design (see lines 159-161). We also included these percentages in Fig. 2.

The authors say this is the first paper to look at the effect of “you” in a bilateral conversational setting. I believe there has been a decent amount of work looking at the effects of pronouns (I, we,

you) in dyadic conversations between romantic partners, particularly in the context of coping etc. The authors may want to qualify or peel back their claim.

Thank you for suggesting this literature to us. We have incorporated this important body of research as we introduce the extant literature and motivate our research. We have also better spelled out how our work is situated in and differentiated from the extant literature in the Introduction (lines 41-50) and Discussion (lines 426-437) sections. In a nutshell, we believe that this work extends the scope of research on pronoun usage in bilateral conversations, from its current focus on close relationships to a broader, non-intimate setting.

*Overall, this is an exciting study and I wish *you* good luck as *you* continue this work :)*

Thank you again for your valuable insights and recommendations, which have been instrumental in advancing our manuscript. Indeed, the integration of the behavioral experiment you suggested has profoundly strengthened the causal claims of the paper. We have also greatly benefited from your other constructive comments, which really helped refine our arguments and offer deeper insights for our readers. We would certainly be happy to address any further comments *you* may share with us as *you* continue to guide us. Thank you again for your precious time and efforts ^ ^

References:

- Heckman, J. J. (1979). Sample selection bias as a specification error. *Econometrica*, 153-161.
- Leung, F. F., Gu, F. F., Li, Y., Zhang, J. Z., & Palmatier, R. W. (2022). Influencer marketing effectiveness. *Journal of Marketing*, 86(6), 93-115.
- Si, K., Li, Y., Ma, C., & Guo, F. (2023). Affiliation bias in peer review and the gender gap. *Research Policy*, 52(7), 104797.
- Su, L., Sengupta, J., Li, Y., & Chen, F. (2023). “Want” versus “Need”: How Linguistic Framing Influences Responses to Crowdfunding Appeals. *Journal of Consumer Research*. Advance online publication.
- Sun, Z., Liu, S., Li, Y., & Ma, C. (2023). Expedited editorial decision in COVID-19 pandemic. *Journal of Informetrics*, 17(1), 101382.

Reviewer #2 (Remarks to the Author):

In this manuscript, the authors studied the effect of using “you” in written communications (i.e., referee report and response of Nature Communications) on the number of questions, number of words, positive/negative embedded in the report. Furthermore, the authors find that using “you” in the response seems to lead a more personal conversation, as it increases subjectivity, and reviewer engagement, but reduces word complexity and first-person singular pronouns usages. Finally, the authors also find that if “you” was initiated by reviewers, the engagement and subjectivity seem significantly stronger. I read the manuscript with great interest. The topic is really interesting, and it contributes to the field of Science of Science and scientists’ behaviors of certain language use. The manuscript is well written. Overall, I believe this manuscript deserves publication after addressing my comments/suggestions below.

We sincerely appreciate your interest in our topic and your positive feedback on the contribution and presentation of our work. We are most grateful for the time and effort you have invested in reviewing our paper. We have diligently followed your suggestions in our revision, with the aim of crafting a paper worthy of publication in *Nature Communications*. In what follows, we quote your original comments in italics and provide our responses to each of them. We hope that you find that our revision efforts meet your expectations.

Major points:

1) The authors divide referee reports according to the usage of “you”, and it seems a binary separation. I am wondering whether a report with MANY “you” shows even less questions/words and more positive comments.

Thank you for your insights, and we agree that it is interesting to investigate whether our effects intensify with increased usage of “you.” In the revised manuscript, we categorize author responses into groups based on the frequency of “you”: a few (one or two), moderate (three through five), and many (six or more) “you” usage. We also designate responses without any “you” as our reference group (SI D, Robustness Check 1). We then re-estimated the benchmark model and reported the results in SI D (Supplementary Table D-1). In a nutshell, we found that the effects of “you” usage on nearly all outcomes amplify as the frequency of “you” rises, save for one variable (i.e., reviewers’ positivity Python package *Textblob*).

For example, recall that our analyses find that authors’ “you” (vs. non-“you”) usage is associated with fewer writings and questions from reviewers in the next round. Here, we find that as authors’ “you” usage increases, the number of words and questions further decreases. Specifically, compared to responses without “you,” “a few” “you” usages see a reduction of 97 words and 2.5 questions; “moderate” “you” usage sees a steeper decrease of 115 words and 2.7 questions; whereas “many” “you” usages see a reduction as much as 211 words and 5.6 questions. While these results do not suggest a strict linear relationship between the frequency of “you” and effect size, they show a general trend: frequent use of “you” appears to be more effective than a simple mention of “you.” We’re grateful for your contribution in highlighting this intriguing perspective in our study.

The manuscript is full of causality words, e.g., lead to. I suggest the authors use these words with caution, as DID doesn't speak to causality and doesn't untangle endogenous issues. Moreover, understanding the underlying mechanisms why such effects exist is necessary. If controlling for subjectivity/work complexity/reviewer engagement, does the main effect shrink?

Thank you for pointing out the issue of causal claims, on which we will elaborate below in detail. To begin, we agree that correlational data and analysis are inherently limited in establishing causality. In line with your suggestion, we have now tuned down the causal language when interpreting the findings from the secondary data (e.g., lines 124, 241, and 261).

In addition, heeding your suggestion to delve deeper into the underlying mechanism, we examined if the impact of “you” usage diminishes when we partial out the effects of variables related to the proposed mechanism, such as subjectivity, word complexity, and reviewer engagement. While there appears to be a reduction in the effect upon controlling for subjectivity, the effect does not significantly decrease when word complexity and reviewer engagement (i.e., LDA topic 11) are accounted for (refer to Exhibit 3 for comprehensive estimation results).

Exhibit 3. The Effect of the “You” Usage When Controlling for Proxies of Mechanisms

(Panel A: Controlling for Subjectivity)

	(1)	(2)	(3)	(4)	(5)	(6)
	Number of Questions	Number of Words	Positivity (Python)	Positivity (R)	Negativity (Python)	Negativity (Hand Coded)
Response with “You”	-3.3135***	-134.0994***	0.0028	0.0166***	-0.0019***	-0.0048***
× After Response	(0.3547)	(14.4546)	(0.0020)	(0.0039)	(0.0007)	(0.0007)
Response with “You”	NA	NA	NA	NA	NA	NA
	NA	NA	NA	NA	NA	NA
After Response	-21.3677***	-1213.2281***	0.0404***	0.0650***	-0.0264***	-0.0316***
	(0.1995)	(8.5916)	(0.0012)	(0.0024)	(0.0004)	(0.0004)
Control Variables	Yes	Yes	Yes	Yes	Yes	Yes
Paper Fixed Effects	Yes	Yes	Yes	Yes	Yes	Yes
Observations	24,640	24,640	24,640	24,640	24,640	24,640
R ²	0.528	0.675	0.431	0.119	0.288	0.416

(Panel B: Controlling for Word Complexity)

	(1)	(2)	(3)	(4)	(5)	(6)
	Number of Questions	Number of Words	Positivity (Python)	Positivity (R)	Negativity (Python)	Negativity (Hand Coded)
Response with “You”	-3.2698***	-133.7545***	0.0063***	0.0166***	-0.0019***	-0.0047***
× After Response	(0.3559)	(14.5471)	(0.0024)	(0.0039)	(0.0007)	(0.0007)
Response with “You”	NA	NA	NA	NA	NA	NA
	NA	NA	NA	NA	NA	NA
After Response	-21.6824***	-1225.9041***	0.0471***	0.0701***	-0.0264***	-0.0321***
	(0.2240)	(9.5532)	(0.0016)	(0.0024)	(0.0004)	(0.0004)
Control Variables	Yes	Yes	Yes	Yes	Yes	Yes
Paper Fixed Effects	Yes	Yes	Yes	Yes	Yes	Yes
Observations	24,640	24,640	24,640	24,640	24,640	24,640
R ²	0.528	0.675	0.431	0.119	0.288	0.416

(Panel C: Controlling for Reviewer Engagement)

	(1)	(2)	(3)	(4)	(5)	(6)
	Number of Questions	Number of Words	Positivity (Python)	Positivity (R)	Negativity (Python)	Negativity (Hand Coded)
Response with “You”	-3.4917***	-150.4695***	0.0077***	0.0168***	-0.0019***	-0.0052***
× After Response	(0.3543)	(14.3375)	(0.0024)	(0.0039)	(0.0007)	(0.0007)
Response with “You”	NA	NA	NA	NA	NA	NA
	NA	NA	NA	NA	NA	NA
After Response	-20.3845***	-1116.1815***	0.0329***	0.0713***	-0.0263***	-0.0296***
	(0.2434)	(11.4619)	(0.0015)	(0.0027)	(0.0005)	(0.0005)
Control Variables	Yes	Yes	Yes	Yes	Yes	Yes
Paper Fixed Effects	Yes	Yes	Yes	Yes	Yes	Yes
Observations	24,640	24,640	24,640	24,640	24,640	24,640
R ²	0.528	0.675	0.431	0.119	0.288	0.416

Our speculation for this result is that intrinsic limitations of the secondary data might have prevented subtle reductions in effects from showing. After all, our analyses employ indirect behavioral proxies (e.g., counting terms like “exciting” or “interesting”) to capture elusive psychological phenomena such as reviewer engagement. These proxies might not be adequately sensitive to detect subtle shift in the proposed mechanism. While we triangulated multiple proxies to demonstrate converging evidence of an engaging and personal conversation, we acknowledge the inherent constraints of secondary data.

Therefore, to provide more direct and conclusive evidence, we have integrated experiments into this revision. We discover that *ceteris paribus*, participants find responses using (vs. not using) second person pronouns to be more positive, and their conversation with the author more personal and engaging (for details see lines 345-350). This result corroborates with all our field data analyses, and provides direct, causal support for our proposed mechanism.

Throughout the manuscript, there are many measurements that are not crystal clear to me, such as word complexity/subjective. I believe some qualitative evidence is necessary. For example, the authors could provide some examples with more subjective sense as well as less subjective sense. OR, some complex words as well as some simple words.

Our apologies for the confusion. In the updated manuscript, we have improved the clarity of the descriptions concerning these measurements in the main text (lines 267-271; 291-292). Also following your suggestion, we have also provided concrete examples of these measures in SI E (Supplementary Table E-1 and E-2). We hope these additions can give our readers a better sense as to what a more (or less) subjective or complex comments could look like in our data.

The authors used LDA to identify topics in order to measure review engagement, and find that reports with “you” seems to show more reviewer engagement. However, it is not clear to me how to measure the engagement, is it the probability to have topic 11? Also, it is great if the authors can provide some test by replacing the relevant topic to unrelated topics.

Thank you for suggesting that we clarify the LDA measurement of engagement. We apologize for placing important details regarding this measurement in the “Method” section towards the end of the paper in the original manuscript. In the updated manuscript, we have elaborated on this measurement both right after the introduction of our engagement topic (topic 11) at lines 320-323, and in the “Method” section at line 529. In a nutshell, reviewer engagement is quantified by the number (frequency) of words associated with the engagement topic. This number is equal to the probability of a review containing the engagement topic (topic 11), multiplied by the total word count of said review.

Following your valuable recommendation, we have also tested our models with the engagement topic (topic 11) replaced by unrelated topics. These unrelated topics, which are used as “placebos,” included topics 13 and 26, which pertain to specific scientific fields (presumably electromagnetism and ecology), and topics 18 and 36, which relate to manuscript evaluation (likely in terms of exposition and methodology). Our findings revealed no significant correlation between the use of “you” and these unrelated topics (detailed results available in SI G (Supplementary Table G-2)). This result further strengthens our confidence in the validity of the proposed engagement mechanism.

For table 4, the authors mentioned “most of the effects become significantly stronger”. I didn’t see empirical evidence, and the authors should elaborate more on this.

We apologize for the confusion created by not having presented the DDD results from the original Table 4 (the new Table 5) in full on our part. In this revision, to clearly show that “most of the effects become significantly stronger,” we split the DDD analysis into two DIDs based on whether the reviewer used “you” in their initial comments. Doing so allows us to estimate the effects of authors’ “you” usage within two different samples: one where the “initial reviews” employ “you” and another where they do not.

We can thus simply contrast the effect sizes of the two respective DIDs (i.e., the coefficients of the DID estimators) to examine reviewer “you” usage’s effect on the outcome variables. As anticipated, the effect sizes of authors’ “you” usage are typically greater when initial reviews used “you” (e.g., $\beta_{\text{engagement}} = 0.1417$) than when they do not (e.g., $\beta_{\text{engagement}} = 0.0613$). Please also refer to lines 369-375 and SI K (Supplementary Table K-1) for more details. The significance of this difference is reported in our formal DDD estimation (Table 5). This estimation indicates the effect of a third dimension of difference (i.e., the heterogeneous reviewer “you” usage) within the original DID analysis. Thank you again for your helpful suggestion.

Minor points:

I suggest to put the description of control variables in the main text, as it is important to know what is the control variable.

Thank you for your recommendation. We have now included the description of the control variables in the main text (see Table 1). Please note that to avoid making the table excessively long in the main text, we chose not to enumerate all levels of the four control variables that have a “fixed-effect flavor” (specifically, the last initial of the first author [26 levels for 26 letters], publication month [12 levels], publication year [6 levels], and paper discipline [5 levels]). A full list of these levels can be found in SI C (Supplementary Table C-1).

In the Tables, it is good to put the coef. of Response with “you”, and After Response as well.

We have now reported the two simple effects, i.e., the coefficients of “Response with ‘you’,” and “After Response” in all tables. Thank you.

If necessary, the authors should also interpret the coef. For example, reports with “you” show XX less words and XX less questions.

Your suggestion is well received! For instance, reviewers addressed by “you” (vs. non-“you”) wrote 135.59 fewer words and posed 3.34 fewer questions on average (see lines 194-197). We concur that this offers more tangible insights for readers. We agree that doing so helps the reader interpret our results in relation to their real-world impact.

Once again, we thank you for your invaluable and constructive suggestions. Your insights have been instrumental in improving the validity of our findings and refining our presentation. We hope that we have addressed your concerns satisfactorily and our revision meets your expectations. Thank you very much again.

Reviewer #3 (Remarks to the Author):

This is a fascinating and quite promising project. The underlying idea is powerful and is something many authors and reviewers have thought about (including me). The sample is remarkable and the methods are generally strong. Despite its considerable strengths, the results are difficult to interpret. To the degree the authors would like to reach a broad audience, I would encourage them (that is, you) to reframe the results with social scientists in mind.

We are deeply grateful that you find our idea powerful and the dataset unique. We are particularly heartened to know that you have thought about the same questions as us, that seemingly mundane and inconspicuous words such as “you” may make a big difference in peer review. Interestingly, some of our additional analyses (to be shown in our response to a related question of yours) seem to suggest that some seasoned, well-published authors might consciously or unconsciously have thought about the same thing too! Greatly encouraged by your insightful advice to reach a wider audience of social scientists, we have closely followed your suggestions to enhance our analytical approach and the presentation of the results.

In the following, we outline our responses to each of your comments. Please note that we have recorded some of your comments and appreciate your kind understanding of that. Doing so allows us to present an overall picture illustrating the cascading of the “you” usage. This big picture is then followed by (1) a detailed explanation of how our “difference-in-difference” (“DID”) model becomes an integral part of the cascading sequence; and (2) potential antecedents of the “you” usage (e.g., certain author characteristics and/or situational factors). In so doing, we aim to offer a clearer and more intuitive depiction of our designs and findings that is friendly to a broader audience. We hope that our revision efforts meet your expectations.

The analytic approach should more closely follow/reflect the social processes you are testing. Using the cascade idea, what are the mean word count (WC) and question count (QC) of authors who are responding to reviews with and without you-words? And then, what are the mean WC and QC of reviewers who respond to authors who do and who do not use you-words? One way to approach the analyses might be something like this:

Level 1: Reviews that Use You-Words Reviews that Don't Use You Words

Level 2: Authors do ... A's don't use ... Authors who do vs don't use You

Level 3: Rev Do/Don't Revs Do/Don't ... Revs Do/Don't Revs Do/Don't

It would be nice to see the actual means either in the text or supplemental information. Adjusted means with the multiple covariates could be used as well. Ultimately, we need to get a sense of the numbers. How big are these effects? The R2 you report undoubtedly include all the covariates. We need to see the specific effects of you-words. Small effect sizes are likely to result but given the question and your use of such glorious real world data, that's OK. Note also that there is an entire industry that is devoted to conversational data that deals with recursive interactions that attempt to disentangle the two sides of a natural conversation (see the important work of DA Kenny, e.g., 2020).

Thank you for this very helpful and constructive comment! Following your cascade idea and suggested analytical approach, we highlight the pattern of two variables of interest (review comments' mean word count [WC] and question count [QC]), following the “you” usage of both the reviewer and the author. SI A (Supplementary Figure A-1) depicts how WC and QC bifurcate following “you” usage and cascade down as the peer review progresses. Note SI A (Supplementary Figure A-1) shows the cascade up to the 3rd round, beyond which point small sample sizes render results less meaningful; This process, however, could theoretically go on into further rounds.

Throughout our responses to you, we employ this cascade of data to address your comments. For more focused demonstrations, most of the time we will focus on a specific section of the entire figure. We have presented these results with sample sizes, as well as the actual mean values of WC and QC (without adjustments from covariates), also following your recommendations to give our readers a sense of how big the effects are.

Here, we first demonstrate how our difference-in-differences (DID) approach, strikingly, aligns almost seamlessly with your cascade idea. We thus maintain that DID can be thought of as an intrinsic part of the cascade process and therefore an appropriate approach for our dataset. To illustrate, first consider the initial branch where 4,344 reviewers began their comments to the authors with “you” [level one]. Subsequently, 2,206 authors responded with “you” and 2,138 did not [level two], before they received reviewer comments in the 2nd round [level three]. Exhibit 4, a section from SI A (Supplementary Figure A-1), further details this branch.

Exhibit 4. An exemplary branch extracted from the full cascading process

At first glance, a mere comparison suggests authors using “you” received more subsequent questions than those not using “you” (8.58 versus 6.77 questions, respectively). This simple comparison, however, can be misleading. This is because authors using “you” and not using “you” may receive different questions in the 1st round to begin with, which happen to be the case here: In our dataset, authors using “you” received more initial questions (37.23) than those not using “you” (32.41). Therefore, the full picture is that, after responding to the reviewers, authors using “you” saw a reduction in the QC by 28.65 (from 37.23 to 8.58), while those not using “you” saw a reduction of 25.64 (from 32.41 to 6.77). In other words, authors using “you” see a greater reduction in QC by the margin of 3.01 (i.e., 28.65 - 25.64), in relative to those not using “you.” By the same token, “you” (vs. non-“you”) usage is associated with a greater reduction of WC by 111.43 words (i.e., [2116.76-674.47] – [1890.21-559.35]).

This analysis on the cascade data shows the very logic of the DID approach: ***the first “differences”*** here are the differences in QC (or WC) before and after author’s response within both the “you” and “non-you” groups, which offset initial discrepancies between the groups. ***The second “difference”*** then contrasts these two differences to estimate the effect of “you” usage — hence the name “difference-in-differences.”

In summary, to accurately identify the effect of “you” usage in cascade data, it is vital to consider the initial variation in QC (or any variable of interest) between “you” and non-“you” groups and its natural progression (e.g., reviewers generally ask fewer questions in subsequent rounds) irrespective of “you” usage. Therefore, DID emerges as the most suitable method to analyze such cascade data. Moreover, this perspective also aids in addressing another related inquiry you posed (discussed below).

Secondary comments

Perhaps the terms of “difference-in-difference” or “DID” and, later, DDD, are common in the authors’ professional specialization. They are not intuitive to an outsider. Why are they called DID? Difference from what? Surely there is a better way to label the phenomenon you are talking about.

Building on your insightful cascading idea, we have now explained how DID works and what its name means. We have also updated our main text in this fashion to explain DID more clearly and intuitively (see lines 58-95). We hope you find our current explanation more friendly to a broader audience. Note that we will explain “DDD” in a later section.

Now, to further build DID into your cascade idea: As shown in Exhibit 5, each branch division, based on the authors’ use of “you,” in fact corresponds to a specific DID. This perspective results in 2 DIDs (highlighted in red in Exhibit 5) in the 1st review round, and 8 DIDs (highlighted in blue in Exhibit 5) in the 2nd round, offering rich opportunities to unravel our data. Encouragingly, a majority of these DIDs corroborate our findings that “you” usage is associated with reduced QC and WC. There are only two reasonable exceptions, wherein the authors did not seem to have fully engaged with the reviewers. For instance, in one such exception (refer to the bottom two 2nd round sub-branches of SI A

(Supplementary Figure A-1)), neither reviewers nor authors employed “you” in any prior exchanges. It was only until this last exchange did the authors use “you,” possibly in a mere “thank you” in response to the reviewers’ acceptance decision. Therefore, there was no further opportunity for engagement.

In conclusion, we have once again shown how DID can be seamlessly integrated into the cascading process. In so doing, we have significantly improved the validity and clarity of our analytical approach. This perspective has also allowed us to garner more insights from data and communicate with a broader audience. We thank you once again for introducing us to this important cascade idea!

Exhibit 5. The embedded DIDs in the cascading process

Notes: We apologize for the size of the figure. For higher resolution, please visit https://osf.io/6azmh/?view_only=f0250d6500d04362963d7132f8fc47e7

We also appreciate your suggestion to include the body of research on dyadic, which explores the interactions and interdependence between two dyad members (e.g., Kenny 2019; Kenny, Kashy, and Cook 2020). We have now included these important works in the Results section (line 380; reference 41 and 42). It is noteworthy, however, that Kenny's Actor-Partner Interdependence Model (APIM) is primarily suited for symmetric dyadic relationships. In contrast, the dyadic relationship between reviewers and authors is relatively asymmetric (or imbalanced), in the sense that reviewers, more than authors, determine key outcomes of the review process (e.g., QC and WC). Nevertheless, motivated by your insights and these important works, we have now explicitly considered the interdependence between two sides of a natural conversation in our revised analytical approach. We will delve deeper into this topic shortly as we address a related question of yours.

Your central argument is that the use of "you" sets up a more personal relationship between the author and reviewer. Until the last paragraph of the results section (lines 285-297), the paper is going on the assumption that the relationship is essentially built by the author in response to the first set of reviews. But if this is a true interaction, isn't it actually set up initially by the first reviews themselves? If I, as a reviewer, talk to you (the author) in a personal way, hasn't the dialog begun? This should not be an afterthought in the paper. It should be part of the cascading interaction of the study: The initial reviews affect the authors' first responses which, in turn, affect the second round of reviewers' comments which then affect the authors' second responses, etc. (In theory, you could start with the original paper but, in reality, virtually no "you" words will be in most papers.)

We apologize for the confusion around who sets up the relationship initially. Now with the help of the cascade process, we are in a more advantageous position to examine this dynamic of relationship and hence this assumption. To do this we build on the two aforementioned DIDs in the first round (i.e., the four first-round branches of Supplementary Figure A-1 in SI A). When viewed together, these two DIDs reveal a ***third difference*** (i.e., whether "you" was used in the initial reviewer comments), thus forming a DDD framework.

To illustrate, let us examine the two DIDs separately. In the DID where initial reviews employed "you," author responses with "you" saw a reduction of 3.01 questions (i.e., [37.23 - 8.58] - [32.41 - 6.77]). Conversely, in the DID without initial "you" usage in review, author responses with "you" saw a reduction of 1.93 questions (i.e., [24.15 - 5.63] - [21.50 - 4.91]). These results illustrate that *regardless* of whether initial reviews used "you" or not, the use of "you" in subsequent author responses is always associated with a reduced number of questions. Therefore, it is not necessary to assume that ***the effect*** of an author's "you" usage is contingent on "you"'s presence in initial reviews. This idea is further supported by the newly added behavioral experiment (see lines 335-361), wherein participants deemed "you" (vs. non-"you") using authors' responses as significantly more engaging, even without "you" in review. In other words, "you" usage has a simple main effect, such that author's "you" usage is already sufficient to influence review outcomes. This also clarifies why our principal analyses focus on the simpler DIDs instead of the more complex DDDs: establishing the simple but powerful effect of "you" is the first-order contribution of this work.

That being said, we totally agree with you that there is also an additive or interactive effect of “you,” wherein mutual, repeated usage is more powerful than unilateral, sporadic usage. Indeed, author “you”’s reduction of QC/WC is more pronounced when initial reviews also use “you” (e.g., 3.01 > 1.93 in the previous example). Moreover, cascading branches with multiple instances of “you” also saw greater QC/WC reductions (SI A: Supplementary Figure A-1). In our formal DDD estimation (Table 5), we have also statistically confirmed the significant impact of “you” in initial reviews, with more control variables.

Note that the discussions above also reflect our new approach to communicating DDD in this revision. Specifically, we split the DDD analysis into two respective yet comparable DIDs based on reviewers’ initial “you” usage (see lines 369-375; SI K: Supplementary Table K-1). This approach allows readers to easily interpret reviewer’s influence on our outcomes, and thus helps us communicate with a broader audience with a greater clarity. Thank you once again for guiding us to think more deeply about the assumption, the presentation of DDD, and the cascading of the proposed effect.

A related question concerns the reliability of you-word usage among multiple reviewers of the same paper. Assume a paper has three reviewers, how frequently do reviewers 1 vs 2 vs 3 use you-words? Similarly, in the authors’ responses to the reviews, do the authors selectively respond to those reviewers who use you-words? What these questions get at concerns the nature of the person who uses you-words. Is there a certain type of personality that tends to use you-words or is this primarily situational? Does a certain type of author or paper provoke you-word use?

Thank you for highlighting concerns regarding potential antecedents of authors’ “you” usage (e.g., author features and/or situational factors), which may lead authors to use “you” to selectively respond to reviewers. To address this concern, we have explored whether an important situation cue (the “you” usage in initial reviews) and two common author features (gender and rank) are predictive of authors’ “you” usage. Here, gender is probabilistically derived from name leveraging the Social Security Administration (SSA) database (Si et al. 2023; Su et al. 2023), and rank is approximated using H-index.

As shown in SI D (Supplementary Table D-6), there is a significant positive association between “you” usage in initial reviews and that in authors’ response. This result reveals reviewers’ initial “you” usage as an important precursor to authors’ subsequent “you” usage. In terms of author attributes, our analysis did not reveal any gender difference. However, we did find that authors with a higher rank (indicated by a greater H-index) are more inclined to use “you.” From an applied perspective, this result suggests that seasoned authors, with an extensive publication history, might instinctively recognize the efficacy of employing “you” in fostering positive and engaging dialogues, and thus opt for more frequent use of second-person pronouns during the review process (This perhaps also helps explain why you thought about the underlying idea before :). For comprehensiveness of analysis, we have also incorporated this selection model with the main DID model (utilizing a Heckman two-stage framework; Heckman 1979; Leung et al. 2023) as an additional robustness check (for details see lines 251-254; SI D: Supplementary Table D-6). The resulting

estimated effect of authors' "you" usage remains consistent, offering further confidence in our findings.

Also note that further evidence suggests that authors do not usually use "you" discriminatively to respond to different reviewers. To test this, we randomly sampled 1,000 papers from our dataset, identified the 347 papers whose authors used "you" to address reviewers, and hand-coded each reviewer's "you" usage for these 347 papers. For each paper, we computed P_{Author} , which is the *proportion* of reviewers the author(s) addressed using "you." For example, $P_{\text{Author}} = 1/3$ means the author(s) addressed 1 out of 3 reviewers using "you." This distribution is presented in Panel (A) of Exhibit 6. Similarly, we also calculated P_{Reviewer} , that is, the proportion of reviewers addressing the author(s) with "you." For instance, $P_{\text{Reviewer}} = 1/3$ means that 1 out of 3 of the paper's reviewers used "you" in their comments. Panel (B) of Exhibit 1 depicts the distribution of P_{Reviewer} .

Exhibit 6 The Distribution of P_{Author} and P_{Reviewer}

Note: x-axis not to scale. Though most papers have 3 reviewers (leading to proportions like 1/3 and 2/3), we also observed proportions like 1/5, 1/4, 1/2, or 3/4, as some papers had 4 or even 5 reviewers.

Particularly notable are the two ends of both figures. To begin, since this dataset includes only those authors who use "you," there are no instances where $P_{\text{Author}} = 0$ (i.e., authors did not address any reviewers using "you"; the leftmost bar in Panel (A)). In contrast, the majority of papers in this dataset (almost 60%) have a $P_{\text{Reviewer}} = 0$, indicating that for most papers, no reviewer addressed the authors using "you," despite the fact that at least one of the reviewers had been addressed with "you" (the leftmost bar in Panel (B)). This divergence indicates that a significant number of reviewers do not reciprocate the "you" usage with authors. This non-matching "you" usage is further corroborated by

the other end of the figures. Indeed, a significant number of papers (over 40%) address all reviewers using “you” ($P_{\text{Author}} = 1$; the rightmost bar in Panel (A)). This indiscriminate usage of “you” cannot result from authors mirroring a specific reviewer’s “you” usage, as very few papers had all their reviewers use “you” in their comments ($P_{\text{Reviewer}} = 1$; the rightmost bar in Panel (B)).

In fact, out of the 347 papers, only about 10% (34) showed an “exact match” in “you” usage – that is, authors used “you” to address only the reviewer(s) who had initially used “you.” Taken together, this sample indicates that while exact “you” matching exists, it is rare at the individual reviewer level (as opposed to the overall paper level).

We hope that these responses have satisfactorily addressed your concerns.

Secondary comments

The LDA analyses are a nice touch. It’s not clear how much they contribute to the main point of the paper however.

Thank you for suggesting that we better clarify the contribution of the LDA analyses. In this revision, we have better elaborated on the purpose and function of this analysis (lines 305-307: 513-518). In a nutshell, the LDA analysis is an attempt to reveal mechanism (i.e., engagement) via a data-driven approach. Specifically, rather than presupposing an “engagement” undertone in the text, the LDA method explores potential latent topics. We then identify a specific topic that happens to be highly associated with the engagement during the review process. We also note that the secondary data has limited ability to capture the underlying mechanism, and hence have conducted an experiment to test underlying mechanism directly and causally (see lines 335-361 for details). The results are consistent across the data- (LDA) and theory-driven (experimental) approaches, lending further confidence in the robustness of our findings.

Line 123. First graphic should include Ns

Thank you for the suggestion. We have now included the number of observations (Ns) in the original Fig. 1 (now Fig. 2).

The tables are difficult to interpret. It’s not clear what columns 1 and 2 represent. I initially thought they referred to “No You Use” and “You Use”. But no. I would like to see the actual mean usage of you-words, either in the main text or supplemental materials.

We apologize for the confusion. In the revised manuscript, we have refined our report of data to better walk readers through our findings. Moreover, we have also included the mean usage of “you” (i.e., 37.74%) both in the main text (lines 159-161) and Fig. 2, following your suggestion. We hope these measures help improve our result reporting.

The courtesy “you” words are quite interesting. Again, it would be helpful to include the means. In other words, what percentage of all you-words are accounted for by courtesy words? To what

degree do courtesy you-words correlate with non-courtesy you-words? If they are correlated at the manuscript level at, say, .30 or higher, it could be argued that they are reflecting the same or similar process (e.g., awareness of the others' perspective as opposed to pure courtesy if the correlation is below .30).

Thank you for the suggestion. Now we have also reported the percentage of courtesy “you” (i.e., 36.63%) in all you-words (line 244). Furthermore, courtesy “you” is significantly correlated with non-courtesy “you” with a correlation coefficient of 0.288 (which is quite close to 0.30). We posit that courtesy “you” might reflect a similar process as non-courtesy “you.” After all, using courteous you, for instance in “thank you” and “we appreciate your help,” can also enhance engagement with reviewers.

In summary, this is a project that has the potential to make a big splash both theoretically and practically. By changing the analytic approach, it will have a far greater impact on researchers in the social sciences.

Jamie Pennebaker

We are deeply grateful for your invaluable ideas and suggestions. In revising the manuscript, our goal has been to enhance its relevance to and impact on researchers in the social sciences. We hope this goal has been achieved in our revamped analytical approach (from simple DIDs to a nuanced cascading perspective), the enrichment of our findings (by exploring the antecedents of authors' “you” usage), and our efforts to improve methodological rigor and clarity in exposition. We sincerely hope our revised work addresses your feedback comprehensively and aligns with your expectations. That said, we will certainly be happy to address any additional comments you may have. We also want to let you know that our entire team has gained significant practical insights from this revision process and considered this one of our most inspiring and enjoyable revision experiences. We deeply appreciate your guidance and time. Thank you once again!

References:

- Heckman, J. J. (1979). Sample selection bias as a specification error. *Econometrica*, 153-161.
- Leung, F. F., Gu, F. F., Li, Y., Zhang, J. Z., & Palmatier, R. W. (2022). Influencer marketing effectiveness. *Journal of Marketing*, 86(6), 93-115.
- Kenny, D. A. (2019). *Interpersonal perception: The foundation of social relationships*. Guilford Publications.
- Kenny, D. A., Kashy, D. A., & Cook, W. L. (2020). *Dyadic data analysis*. Guilford Publications.
- Si, K., Li, Y., Ma, C., & Guo, F. (2023). Affiliation bias in peer review and the gender gap. *Research Policy*, 52(7), 104797.
- Su, L., Sengupta, J., Li, Y., & Chen, F. (2023). “Want” versus “Need”: How Linguistic Framing Influences Responses to Crowdfunding Appeals. *Journal of Consumer Research*. Advance online publication.

Reviewer #4 (Remarks to the Author):

Comments for the author(s):

This paper is, I believe, interesting and important. I am not aware of any other contributions that cover this same kind of area (at least not in the way that this one does), and so it is a novel and significant contribution, I believe. The paper reports noteworthy results focusing on the interactions between authors and reviewers, and in particular the language used in addressing each other with regard to use of the second person, “you”. The findings demonstrate that use of “you” and similar constructions lead to more a positive, personal and engaging conversation between the reviewer and authors.

Your paper is clearly written and does a good job of presenting your data with clarity. I believe that reviewers with particular expertise in the methods you have used will also be commenting on the methodological aspects of the paper in detail, whereas my remarks relate to more general issues, so as not to overreach my expertise. Having said that, the study seems to be well conducted and, as far as I can tell, robust. The paper provides a clear account of the methods used which appear to be sound. The inferences drawn from the data also seem reasonable, although I do have some minor suggestions for how the paper might be improved.

Thank you for your encouraging words on the significance and contribution of our work. Your positive feedback on the clarity, robustness, and delivery of our findings is also deeply appreciated. In this revision, we have closely followed your suggestions, with the aim of crafting a paper worthy of publication in *Nature Communications*. In what follows, we quote your original comments in italics and provide our responses to each of them. We hope that you find that our revision efforts meet your expectations.

I would like to see the review of literature expanded a little, since at the moment it is very brief. This would help provide more context for the study. At present, the second paragraph of the introduction (lines 41-51) contains reference to a number of previous studies (citations 1-7) but there is very little commentary on them and the way in which your study relates to them is only described in a very elliptical way. There is also some reference to previous studies in lines 72-90, although these are presented more in relation to the way you have set up your own study rather than a commentary on the literature as such. The Discussion mentions connections with fields of bibliometrics, scientometrics and metascience, but I would suggest that it would be useful to see these explored upfront to a greater extent.

Thank you for advising us to better situate our research within the existing literature. Following your suggestions, we have summarized the findings of the referenced literature in the Introduction section (lines 41-50). Doing so not only pays homage to previous work, but also reveals uncharted areas in the current literature and hence illustrates how our research advances the field’s understanding of “you” usage. Furthermore, we have improved our Discussion section to better elaborate on how our study contributes to the fields of language study and the Science of Science (lines 426-437; 444-447). We hope these revisions help us better present related research and position our own work.

I would also like to see you define more clearly what you mean (and don't mean) by "subjectivity". In the paper, you use the term in a particular sense to mean more personally engaged language (lines 141-145 indicate that). However, 'subjectivity' can be used in a broader sense, often associated with the idea of personal bias. The Cambridge University online dictionary defines "subjectivity" as "the influence of personal beliefs or feelings, rather than facts". You are not using the term in this broader sense, but it would be useful for you to define your usage more explicitly, and to exclude other connotations.

Thank you for pointing out the need to clarify the term “subjectivity.” Following your suggestion, we have now provided a clear definition of “subjectivity” in the revised manuscript (refer to lines 267-271). Specifically, we adopted the same definition as Bravo et al. (2019), who defined the term as the extent to which a text contains personal opinions as opposed to factual information. Higher subjectivity scores in our data suggest a more opinion-based, subjective reviewer comment, whereas lower scores indicate a more objective stance without many personal opinions. We have also provided concrete examples of more (or less) subjective comments in the SI E (Supplementary Table E-1).

Note that we do not attempt to deny or rule out the potential association between subjectivity and personal bias. Indeed, highly subjective opinions about a paper, such as deeming it “an interesting read,” may very well be the result of inherent biases towards its theme, context, or the methodology employed. Such biases may include, for instance, “I do not believe in field X/ method Y.” In understanding “subjectivity” as a proxy of a personal, engaging conversation between authors and reviewers, we acknowledge that personal biases could, at times, shape this discourse. We hope our revision and explanation help improve the clarity surrounding subjectivity.

I have a number of comments which relate to reviewer use of “you”, as opposed to author use. The focus of your analysis is on author use. Your coverage of reviewers' use of “you” is rather incidental, although you do make some interesting remarks about it from line 275 onwards. However, at other points in your analysis it does feel rather one-sided. I would like to see more about reciprocity between author and reviewer in terms of language, built into the paper more consistently, if your data allows this.

Thank you for encouraging us to delve deeper into the reciprocity between authors and reviewers. As you have correctly pointed out, our DDD analysis (from line 275 onwards in our original manuscript) already touched on this idea. In this revision, to drive home this reviewer-author reciprocity we have conducted and reported additional analyses. To this end, we demonstrate the differential impact of authors' “you” usage, contingent on whether initial reviews also used “you” or not.

Specifically, we split the DDD analysis into two DIDs based on whether the reviewer used “you” in their initial comments. Doing so allows us to estimate the effects of authors' “you” usage within two different samples: one where the “initial reviews” employ “you” and another where they do not (see lines 369-375; SI K: Supplementary Table K-1). Importantly, the effect sizes of authors' “you” usage are typically greater when initial reviews used “you” (e.g., $\beta_{\text{engagement}} = 0.1417$) than when they do not (e.g., $\beta_{\text{engagement}} = 0.0613$). As opposed to a one-sided view, this analysis encompasses “you” usages of

both the reviewer and the author, and demonstrates that reciprocal usage of “you” yields a highly personal and engaging conversation – more so than when only the author employs “you.”

Here, we respond to your comments regarding reviewer-author reciprocity from one angle – specifically, how considering reviewers’ “you” usage affects the outcomes of our original DID analysis (i.e., the *heterogeneous effect* of the “you” usage). In our response to your subsequent related question, we explore yet another dimension of reciprocity in our data. Specifically, we examine how reviewers’ usage of “you” catalyzes authors’ usage of the same – in other words, the *precursor/source* (and subsequently, the reproduction) of the “you” language in the review correspondence.

Your comments on “positivity of review” (from line 170 onwards) raise the question of the origin of “you” language in the review correspondence. You mention author use of “you” as resulting in more positive reviews, but authors are, of course, responding to reviewer comments at that point. You augment this analysis later, from line 275 onwards, with comments about reviewer initiation of “you” language. This does address some of the questions that arise from the earlier analysis. However, I believe the paper would benefit from more analysis of different aspects of author-reviewer reciprocity, where it can be built in. In your Discussion you mention authors following suit in language in responding to reviewers, once again this feels rather one-sided – how does this work the other way round? What are the characteristics of usage that tend to be reproduced in the process of one actor following the suit of the other?

We appreciate your thoughts on exploring language characteristics prone to being reproduced (e.g., following suit). We agree with you that our focus was primarily on how the author influences the reviewers but not the other way around. Following your comments, in this revision we have also highlighted how reviewers influence the author. More specifically, here we examine how reviewer-author reciprocity is pertinent to the origins and reproduction of “you” language.

For clarity, it is beneficial to draw a full picture of the precursors of “you.” Our analyses suggest that the use of “you” (or lack thereof) is influenced by various situational cues and non-situational factors. A key situational factor is the other party’s “you” usage (which we will elaborate below). However, non-situational factors also play a significant role. For instance, authors with higher H-indices tend to use “you” more frequently (see SI D: Supplementary Table D-6). Conversely, around 49% of authors would not use “you” even after being addressed with it. Relatedly, our DID and experiment (lines 335-361) confirm that authors’ “you” usage alone can elicit the effect we found, regardless of the reviewers’ language choice. Overall, while reciprocal “you” usage is influential, it is neither the sole origin nor a necessary condition for the observed effects.

Nonetheless, we maintain that reciprocal “you” usage is an important and intriguing area of study. To this end, we hypothesize that reviewers’ initial “you” language can catalyze its subsequent usage by authors. As SI D (Supplementary Table D-6) indicates, there is a significant positive correlation between “you” usage in initial reviews and that in author responses. Hence, we employed the Heckman two-stage model that allows authors to reciprocate the use of “you” in response to the initial use of “you” by reviewers (lines 251-254; SI D: Supplementary Table D-6). This approach helps us

present a comprehensive view of reviewer-author reciprocity, encompassing both the origin and reproduction of the “you” language.

I would be interested to see more detail on the comment that, "...in our data, authors' usage of "you" is in fact also associated with reviewers' decreased usage of first-person singular pronouns (i.e., "I", "me", "my")". My question again is how does this relate to reviewer use of "you"? You go on to comment on reviewer use of "you" later. Where there is reciprocal use of "you" between authors and reviewers, I would expect use of the first person, "I", to naturally accompany this. As language becomes more personal, the use of the first person might be thought to increase along with the second person. However, this does not seem to be the case in your data and so it would be useful to see this discussed.

Thank you for highlighting this perspective. In our revision, we have taken a closer look at reviewers' use of both singular and plural first-person pronouns (see SI F (Supplementary Table F-1) for detailed estimation results). The findings indicate that when authors use “you” (vs. non-“you”), reviewers use significantly fewer singular first-person pronouns. While we also agree with your hunch that people may naturally use “I” to go with “you,” “I” usage is often associated with self-focus, egocentrism, or even narcissism, which are antithetical to an engaging, mutual conversation. Indeed, some research has shown that the use of “I” is sometimes associated with negative emotions and dissatisfaction in romantic partnerships (Seider et al., 2009; Sillars et al., 1997).

Conversely, reviewers' use of plural first-person pronouns remained relatively stable, regardless of authors' “you” usage. The interpretation of this null effect is beyond the scope of this research and is not supported by our current data and analysis. Nonetheless, one may refer to extant literature for potential insights. For example, “we” language is widely recognized to reflect a communal orientation in close relationships (Seider et al. 2009; Williams-Baucom et al. 2010) but often hinges on a shared identity to be effective (Karan, Rosenthal, and Robbins 2019).

We have now brought this intriguing discussion to readers (lines 285-289). Thank you again for the excellent suggestion.

A final point related to this is that of confrontational use of “you” and “I”. Sometimes in confrontational situations the use of “you” versus “I” can be seen as heightening tension (see Rogers, S. L., Howieson, J., & Neame, C. (2018). I understand you feel that way, but I feel this way: The benefits of I-language and communicating perspective during conflict. PeerJ, 6, e4831. <https://doi.org/10.7717/peerj.4831>) Do you see any evidence of this or have any comments on it in relation to your findings?

Thank you for introducing us to Rogers, Howieson, and Neame (2018), who delved into the confrontational implications of “you” in communication. We have now included this important work and the idea of confrontational “you” usage in the revised manuscript as part of our Discussion (line 455; reference 62). Recognizing the limitations of secondary data in identifying confrontational nuances, we explored this potential mechanism through an experimental approach (see SI I and J). In this study, we manipulated “you” usage (“you” vs. non-“you”), and measured perceived

contentiousness of the conversation (see lines 468-474 in SI J for details). We find no association between “you” usage with contentiousness, nor does including contentiousness as a covariate influence “you”’s effect on positivity (for details see lines 486-487 in SI J). This result shows that the effect of “you” usage on positivity and engagement in a personal conversation persists even when contention is taken into account.

That said, we acknowledge that “you” can carry contentious undertones in various communicative contexts – even within our dataset. The question then arises: why is the contentious effect of “you” not evident in our findings, when it might counteract the positive effects we observed? Although we cannot pinpoint instances of confrontational “you” usage in our data, we speculate that for prestigious journals such as *Nature Communications*, the peer review dataset might predominantly showcase experienced reviewers and authors engaging in constructive discussions around high-caliber work. It is thus plausible that within the present dataset, contentious “you” usage is overshadowed by its positive and collaborative counterpart. As we elaborate in the Discussion section, to better uncover the confrontational aspects of “you” usage, alternative datasets might be required – perhaps those containing rejected submissions or capturing inherently more contentious communications.

I would be very interested to see examples of sentences use the illustrate your key points, even if in appendices. As it is, the focus of reporting is on individual words rather than on sentences, and whilst I realise this is a consequence of the way in which you have conducted your analysis, it would be useful to have some example sentences included.

The supplementary material is helpful and clear, although I am not sure of the value of the word cloud which, if it comes from all of your data, doesn't really tell us much.

We appreciate the two insightful suggestions of yours, and would like to address them together. First – yes, the word cloud of subjective words comes from all of our data. While terms like “interesting” within the word cloud may give our reader a sense of “subjectivity,” we agree that more contextualized, concrete examples for key terms can help enhance understanding. Accordingly, following your recommendation, we have added a selection of sample sentences to the SI E (Supplementary Tables E-1 and E-2). We hope these examples will give readers a clear grasp of more (or less) subjective or complex sentences found in reviews. By presenting these words within full sentences, we believe we have communicated our key concepts more effectively.

The writing of the article is clear and it is very well presented. There are a few minor proofing areas that need correcting.

Thank you for your kind comments regarding the clarity and presentation of our work. In response to your suggestion, we have also addressed minor proofing areas in this revision. As we conclude, we want to express our deepest gratitude for your encouraging comments and invaluable guidance. We also thank you for your valuable time dedicated to reviewing our work. We hope our revisions have met your expectations and have adequately addressed your concerns.

References:

Bravo, G., Grimaldo, F., López-Iñesta, E., Mehmani, B., & Squazzoni, F. (2019). The effect of publishing peer review reports on referee behavior in five scholarly journals. *Nature Communications*, 10(1), 322.

Karan A., Rosenthal R., Robbins M. L. (2019) Meta-analytic evidence that we-talk predicts relationship and personal functioning in romantic couples. *Journal of Social and Personal Relationships* 36(9), 2624–2651.

Seider B. H., Hirschberger G., Nelson K. L., Levenson R. W. (2009) We can work it out: Age differences in relational pronouns, physiology, and behavior in marital conflict. *Psychology of Aging* 24(3), 604–613.

Si, K., Li, Y., Ma, C., & Guo, F. (2023). Affiliation bias in peer review and the gender gap. *Research Policy*, 52(7), 104797.

Sillars A., Shellen W., McIntosh A., Pomegranate M. (1997) Relational characteristics of language: Elaboration and differentiation in marital conversations. *Western Journal of Communication* 61(4), 403–422.

Su, L., Sengupta, J., Li, Y., & Chen, F. (2023). “Want” versus “Need”: How Linguistic Framing Influences Responses to Crowdfunding Appeals. *Journal of Consumer Research*. Advance online publication.

Williams-Baucom K. J., Atkins D. C., Sevier M., Eldridge K. A., Christensen A. (2010) “You” and “I” need to talk about “us”: Linguistic patterns in marital interactions. *Personal Relationship*. 17(1), 41–56.

Reviewer #5 (Remarks to the Author):

Thank you for the opportunity to review “Two can play at that game of ‘you’: The behavioral consequences of using second-person pronouns in written communications”. This paper uses peer review correspondence collected from Nature Communications to examine the relationship between authors’ use of second-person pronouns in their initial response to reviewers and various characteristics of reviewer responses in subsequent rounds of peer review. In particular, the authors utilize a differences-in-differences (DID) approach in attempt to identify the causal effect of second-person pronoun usage on the structure, sentiment, style of reviewer responses. While the paper tackles an interesting question about the nature of author-reviewer interactions in the peer review process, a key weakness of the paper lies in its statistical methodology, which is, in many places, weak and/or inadequately explained. For that reason, I believe the manuscript would benefit from a major revision.

Some general feedback as well as several specific comments are included below.

Thank you for your interest in our topic. We deeply appreciate your insightful comments regarding the statistical methods. In what follows, we quote your original comments in italics and provide our responses to each of them. We hope that you find that our revision efforts meet your expectations.

General

• Throughout the paper, the authors utilize a DID approach to examine the effect of “you” versus non-“you” usage on aspects of reviewer correspondence. While I agree that this is an appropriate model for the problem at hand, the authors fail to acknowledge any of the (strong) assumptions required for inference in this setting and offer no discussion of the methodological limitations of their analysis. Without addressing these critical issues, I am skeptical about the credibility of the paper’s claims, particularly those regarding causality.

Thank you for your positive feedback on the suitability of the DID approach. We apologize for not having adequately addressed the assumptions and methodological limitations in our original work. In the Discussion section of the revised manuscript (lines 448-452), we acknowledge that the data may have limitations in verifying the equal trends assumptions directly (i.e., no significant time-varying differences between the treatment and control groups in the outcomes). This is due to the fact that there were no historical reviews available before the review process commenced for the analyzed papers. Consequently, the credibility of the causal claims largely relies on whether the treatment of the “you” usage is sufficiently random (or exogenous). To this end, we have strengthened the methodological approach and conducted thorough checks to mitigate the concern of non-randomness (confounding) of authors’ “you” usage, on which we will elaborate in detail in our response to your upcoming question.

Additionally, following the Senior Editor and review team’s suggestions, we believe that a controlled *behavioral experiment* is the most effective way to tackle concerns over potential confounding in authors’ usage of “you.” Specifically, the experiment allows us to draw causal inferences by testing the proposed effects (and underlying mechanism) in a strictly controlled setting that isolates the focal

effect (i.e., that of “you” usage) from confounding factors. In a nutshell, we conducted a behavioral experiment (N = 1,601) in which participants, assuming the role of reviewers, were presented with identical authors’ responses except for the use of second- versus third person- (e.g., “As per your [the reviewer’s] recommendation”). We discover that *ceteris paribus*, participants find responses using (vs. not using) second-person pronouns to be more positive, and their conversation with the author more personal and engaging (for details see lines 345-350). This result corroborates with all our large-scale field data analyses, and provides direct, causal support for our proposed mechanism. Thank you again for encouraging us to strengthen the causal claims of our work.

Related to the issue above, the authors define the intervention (here, the use of 2nd person pronouns in an authors’ first round of responses to reviewer comments) in a manner that allows units to self-select into the treatment and control groups. As a result, there is tremendous potential for confounding due to pre-existing differences between groups. For instance, perhaps authors who receive generally positive feedback in their first round of reviewer comments are more likely to use “you” in their responses compared to those who receive more critical feedback. Similarly, it seems that an authors overall writing style or the strength of their arguments could explain the relationship between pronoun usage and aspects of reviewer responses. While I don’t believe these issues are necessarily insurmountable, they do enough of a philosophical threat in the present analysis to warrant consideration.

Thank you for your attention to this matter. We have taken your comments to heart and have also enhanced the empirical analyses in addition to conducting the experiment. Specifically, we enhanced the empirical identification and conducted a series of robustness checks to mitigate potential confounding factors and ensure the reliability of our findings:

To begin, we have extended the range of control variables (see lines 547-551; Table 1) to include, as you kindly suggested, (1) the positivity of 1st round reviewer comments; and (2) author-related characteristics, such as the positivity and friendliness of authors’ response, as well as the gender and rank of the authors. For completeness of analysis, we have further attempted to include author fixed effects into the model and the results remained stable and robust (see Exhibit 7 below). However, we advise caution when introducing author fixed effects, since 88% of the authors published only one paper in *Nature Communications*. This creates high collinearity between author and paper fixed effects. Therefore, it is more parsimonious to include only the paper fixed effects.

Exhibit 7. The Effect of the “You” Usage When Controlling for Proxies of Mechanisms

	(1)	(2)	(3)	(4)	(5)	(6)
	Number of Questions	Number of Words	Positivity (Python)	Positivity (R)	Negativity (Python)	Negativity (Hand Coded)
Response with “You”	-3.3360***	-135.5879***	0.0051	0.0173***	-0.0019**	-0.0048***
× After Response	(0.4967)	(20.2659)	(0.0034)	(0.0054)	(0.0009)	(0.0010)
Control Variables	Yes	Yes	Yes	Yes	Yes	Yes
Paper Fixed Effects	Yes	Yes	Yes	Yes	Yes	Yes
Author Fixed Effects	Yes	Yes	Yes	Yes	Yes	Yes
Observations	24,640	24,640	24,640	24,640	24,640	24,640

In addition, we have conducted several robustness checks to alleviate concerns over potential non-randomness (exogeneity) of authors’ “you” usage. First, we have combined the DID approach with matching techniques – specifically, propensity score matching (PSM) – to obtain “matched” experiment and control groups that possess comparable observable characteristics (lines 248-251). Consistent results are obtained through the use of the PSM-DID approach (SI D: Supplementary Table D-5). Second, we have employed Heckman-type model to explicitly account for authors’ potential selective use of “you” (lines 251-254). In particular, the selection model has taken into account the reviewers’ initial use of “you,” which can prompt authors’ response with “you” (i.e., the relevance criterion) but may not directly influence reviewers’ behavioral outcomes in different revision rounds (e.g., the exclusion restriction). Once again, the estimates of this alternative model are consistent with our findings (SI D: Supplementary Table D-6). Finally, to cater for non-randomness due to unobserved variables, we have conducted a placebo test (lines 254-257) wherein the treatment group is randomly generated (allowing replacement, while retaining group size unchanged as 38% of the papers). We then used the generated “placebo” treatments (of the “you” usage) to re-estimate the DID model, stored results, and repeated this process 500 times. As displayed in SI D (Supplementary Figure D-2), the distribution of the estimates (of the placebo “you” usage) is centered around zero, and as expected, our benchmark estimates clearly lie outside the range of the placebo estimates.

We hope that these extensive controls and robustness checks have enhanced your confidence in our conclusions.

Much of the Discussion section presents background information that connects the present analysis to existing literature. As a result, this section reads more like a second Introduction than a typical discussion. I would recommend restructuring this part of the paper to include a more detailed examination of the paper’s main findings as well as a discussion of the paper’s strengths and weaknesses, with a particular emphasis on the methodological limitations.

Thank you for the helpful comments. Following your suggestion, we have revised the Discussion section, providing a more comprehensive discussion of the paper’s main findings. In particular, we emphasized (1) constraints of secondary data in confirming the equal trends assumptions; as well as (2) constraints of our methodology in tackling non-randomness issues related to authors’ “you” usage,

and in inferring psychological mechanism. Additionally, we have also highlighted the strength of the paper in its incorporation of behavioral experiments. This approach allows us to test our theory directly and in a controlled setting, reinforcing our causal claims and enhancing the findings' validity.

Specific

On line 112 the authors state that their data includes “13,359 published papers that account for a total of 29,144 round of review”. However, in Table S1, the authors present summary statistics for a sample of N = 26, 679 documents and in the majority of their analyses they list a sample size of N = 26, 640. Where are these discrepancies coming from?

Related to the above, did all papers included in the analysis make it through at least 2 rounds of review before acceptance? If not would be helpful to present the total number of papers represented in each phase of the review process.

Thank you for your comments and suggestions! We agree that these questions around numbers of papers, review rounds, and DID sample sizes are very important for the clarity of our manuscript. Following your suggestion, please first refer to Exhibit 8 below, which presents the total number of papers in each round of review.

Exhibit 8. Total number of papers in each round of review

Review round	Total number of papers
1	13,359
2	12,320
3	3,320
4	102
5	43

Now based on this exhibit, we address your questions regarding discrepancies in reported observations, as well as whether all papers included in the analysis made it through at least 2 rounds of review before acceptance:

1. As shown in Exhibit 8 and explained in the “Data and Design” subsection under the “Results” section (lines 135-136), our initial, full data set includes 13,359 papers.
2. As illustrated in Fig. 2 of the main text, all 13,359 papers were included in the DID model. Specifically,
 - a. For the 12,320 papers (see Exhibit 8) that underwent *at least two rounds* of review, the DID model includes their reviewer comments from the 1st and 2nd rounds.
 - b. For the 1,039 papers that were accepted after *only one round* of review, the DID model includes only the reviewer comments from the 1st round (because the 2nd round reviewer comments do not exist).
3. Our DID model thus begins with a total of 25,679 observations of reviewer comments (12,320 papers × 2 rounds per paper + 1,039 papers × 1 round per paper). Our “basic” DID

estimations (i.e., without fixed effects and other controls) are conducted based on this sample size (i.e., Columns (1) and (3) in Table 2 of the main text).

4. When paper fixed effects are included, the DID model further drops singleton observations that arise from these 1,039 accepted papers, leading to a final sample size of 24,640 observations (25,679 - 1,039). All remaining DID estimations reported in the manuscript are based on these 24,640 observations.

In a nutshell, the discrepancies you spotted are a result of the acceptance of 1,039 papers after the 1st round of review. These 1,039 papers were initially included in the DID model, but later dropped from analysis as singleton observations once the fixed effects and other controls were included in the model. For your and other readers' easy reference, the clarifications above are now included in SI 2.

We hope our response helps clarify the issues around sample sizes and rounds of reviews, and thank you again for helping us improve our data reporting!

On lines 118-122, the authors state that they focus on the first review-response-review process for their analysis. However, Fig. 1 shows multiple rounds of review. It is unclear to me whether the text comments from rounds 2 and after used for any part of this analysis. If not, the authors might consider updating or removing Fig. 1 to avoid confusion.

We apologize for the confusion caused. Indeed, the first review-response-review process includes reviews from both the 1st and 2nd rounds, allowing us to calculate temporal changes in our variables of interest (e.g., positivity, engagement). Following your advice, we have updated Fig. 1 by removing all unnecessary information (e.g., 3rd round reviews and onwards) and highlighted the sample used in the analysis (i.e., reviews before and after author's 1st round response). Please also note that we have chosen to retain Fig. 1, as it offers readers, especially those from the social sciences who might be unfamiliar with DID methodology, a clear visual representation of our analysis method. We appreciate your kind understanding.

In Table 1, what is the purpose of including the "basic DID regression" without controls or fixed effects for these particular outcomes?

Thank you for raising this point. We understand that the formal setup for DID typically includes controls and fixed effects. Here, we also provided the "basic" DID (i.e., without controls and fixed effects) to present preliminary support for the proposed effects, and to disclose if the effect sizes remained relatively consistent after we introduced controls and fixed effects. It turns out they did. We hope our inclusion of the basic DID strengthens the reliability of the conclusions.

On line 251, when the authors state that they employ "LDA on our corpus to identify topics relevant to the reviewers' engagement", I am confused about what corpus they are referring to. Was LDA run on only the first round of reviewer comments? Only the second round? Some combination?

For completeness of analysis, we have applied LDA on the corpus that consists of the first and second round review comments (lines 307-308). Thank you for the comment.

For the LDA analysis beginning on line 250:

– I assume the topic model is estimated using reviewer comments from both the treatment and control group. Since: 1) there are considerably more observations in control and 2) reviewer comments tend to be longer in the control group, the estimated topic model is likely to be dominated by language that may be specific to the control group. A structural topic model might be more appropriate to address this issue.

Thank you for encouraging us to delve deeper into the LDA analysis. First, yes, the topic model is estimated using both the treatment and control groups. We concur that the control group may have a greater impact on the determination of topics. Following your suggestion, we have alternatively constructed a structural topic model (STM) which considers the source of a review comment (treatment or control group) as a factor in the model's estimation. SI G (Supplementary Table G-1, Column (4)) shows that, although the estimated effect is not as substantial ($p = .168$) as in the original LDA model, the direction of the effect remains the same, which overall aligns with proposed account.

We would also advise caution when placing significant reliance on the STM. Note that the covariate (treatment versus control group) is already intertwined in the generation of the engagement topic in the STM. Therefore, when regressing the engagement topic against this covariate in the subsequent DID model, one appears to impose a correlation between the engagement and the covariate from the outset. Having said that, we completely agree that this method provides a viable alternative approach to assess the robustness of our proposed account.

It would be helpful to show the distribution of document-level topic proportions within the chosen topic. Is prevalence within this topic generally high or low, and with how much variability?

Thank you for the advice! We have now shown the distribution of the engagement topic's percentage across the documents (SI G: Supplementary Figure G-3). The low prevalence of the engagement topic (1.1%) is to be expected: After all, there are as many as 40 topics in the text; Moreover, the majority of said text is more likely to be languages oriented towards substantial matters of the research than those employed to engage with people. On the other hand, the distribution reveals significant diversity in the topic, as evidenced by a standard deviation of 2.5% and a maximum value of 41.9%. These variabilities provide ample opportunities for discerning the impact of the "you" usage.

The authors seem to make a very strong claim that: 1) their topic model is stable, and 2) the estimated topic proportions for their identified topic are an accurate reflection of reviewer engagement. Is there any precedent in the literature to support this? Why not test for differential use of a subset of "high-engagement" words (e.g., based on word counts), which would avoid the complications described above?

Thank you for your valuable suggestion. Yes – we do attempt to utilize a relatively stable topic model to explore the mechanism. That said, we also recognize that our LDA approach is more of an exploratory than a confirmatory approach to this effort. Indeed, to derive the engagement topic, we examine word lists associated with each topic and identify one that suggests engagement, rather than definitively confirming it. Although it adds a layer of complexity to our main story, the LDA approach offers readers a data-driven perspective, illustrating how engagement topics might naturally and organically emerge in reviews (e.g., your “revision” “addressed” my “concern”).

We also agree that using predetermined engagement words, as an alternative to the LDA-generated ones, certainly serves as a valuable complementary approach. Following your recommendation, we manually compiled a list of engagement words (SI H: Supplementary Table H-1) based on word counts. We then employed this set to reassess the impact of the “you” usage on engagement. The results once again support a positive association between the “you” usage and engagement (SI G: Supplementary Table G-1, Column (3)), albeit with slightly reduced significance. We suspect that this diminished significance might stem from the challenge in formulating a predetermined (as opposed to data-driven) word list that effectively captures spontaneous, real-world engagement. For example, while a predetermined word list might use descriptive terms like “engage” or “collaborate,” organic conversations often involve context-specific phrases like “your revision addressed my concern” or “following your recommendations.”

As is inherent with observational studies, the mechanistic insights one can glean from secondary data are often limited. To that end, we have conducted a behavioral experiment that allows us to draw direct mechanistic insights (see lines 335-361; SI I and J for details), as explained in our response to your general questions. Therefore, while the LDA analysis offers but an initial exploration into the mechanism, the confirmatory evidence from our experiment affirms the proposed mechanism’s validity. Once again, we deeply appreciate your insightful feedback.

On line 290, in what sense do the effects become “significantly stronger”? Is this statement based on the magnitude of the estimated effects, overall model fit, or something else?

We apologize for the confusion due to not having presented the DDD results around the original line 290 in full. Yes, the comparisons are made based on the magnitude of the estimated effects. To clearly elaborate that the effects become “significantly stronger,” in this revision we have split the DDD analysis into two respective DIDs wherein the effects of authors’ “you” usage are separately estimated in two samples: one in which initial reviews use “you,” and the other one where they do not (see lines 369-375; SI K: Supplementary Table K-1).

In line with our expectation, the effect sizes of authors’ “you” usage are generally larger in the sample of you-based “initial reviews” (e.g., $\beta_{\text{engagement}} = 0.1417$) versus non-you-based “initial reviews” (e.g., $\beta_{\text{engagement}} = 0.0613$), clearly illustrating a stronger effect of “you” usage when “initial reviews” also use (versus do not use) you. While presenting two separate DIDs provides more details, the overall DDD analysis has the advantage of leveraging full data and directly testing the significance of the heterogeneous effects of the “you” usage (arising from “initial reviews”). Together, they provide a full picture to the readers. Thank you again for your helpful suggestion.

Equation (2) should include a plus sign at the end of line 421

Thank you for catching that! The plus sign (line 539) has now been restored.

In closing, thank you for your insightful and constructive feedback. Your comments have been instrumental in enhancing the validity of our findings and strengthening our causal assertions. We hope that we have addressed your concerns satisfactorily and will certainly be happy to address any additional comments you may have. Thank you again for your time and guidance.

REVIEWERS' COMMENTS

Reviewer #1 (Remarks to the Author):

The authors have been extremely responsive by fully addressing all of my comments.

I was particularly encouraged to see the experimental results, which bolster the findings from the field experiment. Further, your direct replication of your experiment and the mediation findings further strengthen your claims. The literature that you integrated and the additional analyses (e.g., inclusion of additional control variables, inspection of other conversational dynamics--like who initiated the "you" usage) also strengthen the paper. Overall, this paper offers a straightforward hypothesis, which is tricky to study. The authors test their hypothesis using a fantastic dataset and creative methodological and analytic approaches. I believe it will make a nice contribution to the literature and I look forward to the future investigations these findings will encourage.

Reviewer #2 (Remarks to the Author):

The authors have successfully addressed my comments, and I appreciate their efforts. The manuscript is much clearer, and I support its publication in Nature Communications once the authors address the following minor issue.

As the main sample of the paper consists of articles that are successfully published, it is helpful to mention possible selection issues in the limitation section.

Beyond this point, I have no further comment. Congratulations to the interesting work.

Reviewer #3 (Remarks to the Author):

This has been an enjoyable and quite interesting process to be a part of this review. I continue to be enthusiastic about the paper and strongly support its publication. I've rarely seen such positive and lengthy initial reviews to a paper. By the same token, it is rare to find reviews so responsive. In addition to changes in the paper itself, the addition of the two experiments in the SI is a nice touch.

In response to the standard questions:

The results are noteworthy and are summarized in the abstract.

This paper will be of interest to researchers across disciplines but especially to those with interests in natural language.

The paper does a fine job in supporting its conclusions.

There are no flaws that would require revisions.

The methodology is sound.

There is enough detail.

The only minor point that that I would quibble about is the discussion of first person singular pronouns, or I-words. There are now a very large number of studies looking at I-words across multiple contexts including natural conversations, more formal interactions, speeches, social media, etc. The evidence is quite strong that I-words are associated with lower status, being personal, anxious, depressed, humble, lower self-esteem, insecure, poorer, and younger. Contrary to common belief, it is not consistently related to narcissism, or positively correlated with egocentricity, being male, or high self-esteem.

Typically, an increase in I-words is linked to being more personal. And this is where your study is noteworthy. It looks as if the reviewers' language continues to remain low in I-words. What is important to appreciate is that there are rarely any times where there is such a glaring disparity in status than between an author and reviewer. The author always has to walk on eggshells and the reviewer can hide behind the wall and make pronouncements anonymously. A high status reviewer can appear to be quite warm and authentic without using I-words – and that's what I think your raters see.

I would have predicted that reviewers who are positively disposed to the paper would use more I-words but it's harder to predict how this would change from the first to the second review. Whether or not you address this point in the paper is up to you. It is a minor point that interests me and perhaps few others.

Jamie Pennebaker

Reviewer #4 (Remarks to the Author):

The revised version of the paper seems to have adequately addressed my comments, as explained in the authors' response. I am happy to recommend the paper for publication.

Reviewer #5 (Remarks to the Author):

Thank you for the opportunity to review the revised manuscript for "Two can play at that game of 'you': The behavioral consequences of using second-person pronouns in written communications". It is clear that the authors have made a considerable effort to address reviewer feedback, and I believe the paper has improved considerably as a result.

The authors have satisfactorily addressed my main concerns regarding the credibility of the causal claims and limitations of the methodology. In particular, the authors have conducted extensive robustness checks using propensity score matching, Heckman models, and placebo tests, which provide reassuring evidence that the effects are not driven by confounding factors. The experimental study also provides direct support for the proposed mechanism. Together, these additions reinforce the validity of the findings and strengthen the credibility of the work.

I also appreciate the authors' attentiveness to clarifying details about the sample, methodology, and results presentation. The additional information they provide about the number of papers in each round of review, discrepancies in sample sizes, and specifications for the DID & DDD analysis add needed transparency to the manuscript.

Overall, I believe the authors have been highly responsive to my previous comments, and their revisions show a commendable level of rigor. The revised paper makes a compelling case for the impact of second-person pronoun use and sheds important light on the dynamics of peer review correspondence. I have no major concerns regarding publication and am enthusiastic that this study will make an important contribution to the literature.

Point-by-point response to reviewer comments

Reviewer #1 (Remarks to the Author):

The authors have been extremely responsive by fully addressing all of my comments. I was particularly encouraged to see the experimental results, which bolster the findings from the field experiment. Further, your direct replication of your experiment and the mediation findings further strengthen your claims. The literature that you integrated and the additional analyses (e.g., inclusion of additional control variables, inspection of other conversational dynamics--like who initiated the "you" usage) also strengthen the paper. Overall, this paper offers a straightforward hypothesis, which is tricky to study. The authors test their hypothesis using a fantastic dataset and creative methodological and analytic approaches. I believe it will make a nice contribution to the literature and I look forward to the future investigations these findings will encourage.

We are deeply humbled by your kind words and encouraging feedback. We are also most grateful for your insights from the previous round, which proved invaluable to our work's development. We, too, are eager to see future research developments in this literature and are grateful for the opportunity to contribute to this collective endeavor.

Reviewer #2 (Remarks to the Author):

The authors have successfully addressed my comments, and I appreciate their efforts. The manuscript is much clearer, and I support its publication in Nature Communications once the authors address the following minor issue.

As the main sample of the paper consists of articles that are successfully published, it is helpful to mention possible selection issues in the limitation section.

Beyond this point, I have no further comment. Congratulations to the interesting work.

Thank you very much for your support, and we are greatly encouraged to know that we have successfully addressed your comments! In response to your remaining minor comment, we have included a discussion in our manuscript regarding possible selection bias in our dataset, stemming from the inclusion of only successfully published works (lines 382-390).

Reviewer #3 (Remarks to the Author):

This has been an enjoyable and quite interesting process to be a part of this review. I continue to be enthusiastic about the paper and strongly support its publication. I've rarely seen such positive and lengthy initial reviews to a paper. By the same token, it is rare to find reviews so

responsive. In addition to changes in the paper itself, the addition of the two experiments in the SI is a nice touch.

In response to the standard questions:

The results are noteworthy and are summarized in the abstract.

This paper will be of interest to researchers across disciplines but especially to those with interests in natural language.

The paper does a fine job in supporting its conclusions.

There are no flaws that would require revisions.

The methodology is sound.

There is enough detail.

The only minor point that that I would quibble about is the discussion of first person singular pronouns, or I-words. There are now a very large number of studies looking at I-words across multiple contexts including natural conversations, more formal interactions, speeches, social media, etc. The evidence is quite strong that I-words are associated with lower status, being personal, anxious, depressed, humble, lower self-esteem, insecure, poorer, and younger. Contrary to common belief, it is not consistently related to narcissism, or positively correlated with egocentricity, being male, or high self-esteem.

Typically, an increase in I-words is linked to being more personal. And this is where your study is noteworthy. It looks as if the reviewers' language continues to remain low in I-words. What is important to appreciate is that there are rarely any times where there is such a glaring disparity in status than between an author and reviewer. The author always has to walk on eggshells and the reviewer can hide behind the wall and make pronouncements anonymously. A high status reviewer can appear to be quite warm and authentic without using I-words – and that's what I think your raters see.

I would have predicted that reviewers who are positively disposed to the paper would use more I-words but it's harder to predict how this would change from the first to the second review.

Whether or not you address this point in the paper is up to you. It is a minor point that interests me and perhaps few others.

Jamie Pennebaker

We are very delighted to know that you found the review process enjoyable and interesting. Your enthusiasm and support for our work are much appreciated! Regarding the minor point you raised, we agree that the psychological underpinnings of “I” usage are indeed rich and nuanced, and our context adds an extra layer of complexity with a striking power difference between parties. We have now included a discussion in the manuscript, acknowledging the alternative theoretical accounts and competing empirical findings in interpreting “I” usage (lines 392-397). With the public release of our dataset, we also call on future research to explore the use of “I” in written communications more rigorously.

Reviewer #4 (Remarks to the Author):

The revised version of the paper seems to have adequately addressed my comments, as explained in the authors' response. I am happy to recommend the paper for publication.

We greatly appreciate your recommendation for our work's publication! We would also like to express our appreciation for your insightful comments in the previous round, which substantially improved our work.

Reviewer #5 (Remarks to the Author):

Thank you for the opportunity to review the revised manuscript for "Two can play at that game of 'you': The behavioral consequences of using second-person pronouns in written communications". It is clear that the authors have made a considerable effort to address reviewer feedback, and I believe the paper has improved considerably as a result.

The authors have satisfactorily addressed my main concerns regarding the credibility of the causal claims and limitations of the methodology. In particular, the authors have conducted extensive robustness checks using propensity score matching, Heckman models, and placebo tests, which provide reassuring evidence that the effects are not driven by confounding factors. The experimental study also provides direct support for the proposed mechanism. Together, these additions reinforce the validity of the findings and strengthen the credibility of the work.

I also appreciate the authors' attentiveness to clarifying details about the sample, methodology, and results presentation. The additional information they provide about the number of papers in each round of review, discrepancies in sample sizes, and specifications for the DID & DDD analysis add needed transparency to the manuscript.

Overall, I believe the authors have been highly responsive to my previous comments, and their revisions show a commendable level of rigor. The revised paper makes a compelling case for the impact of second-person pronoun use and sheds important light on the dynamics of peer review correspondence. I have no major concerns regarding publication and am enthusiastic that this study will make an important contribution to the literature.

Thank you for your encouraging words and valuable feedback! Your insights have been indispensable in enhancing both the theoretical and methodological rigor of our study, and we sincerely appreciate the time and effort you dedicated to reviewing our work.